# What's Wrong with Deep Learning in Tree Search for Combinatorial Optimization

**Maximilian Böther,**[*] **Otto Kißig & Martin Taraz**
Hasso Plattner Institute, University of Potsdam

**Sarel Cohen**
The Academic College of Tel Aviv-Yaffo, Israel

**Karen Seidel**
Department of Mathematics, University of Potsdam

**Tobias Friedrich**
Hasso Plattner Institute, University of Potsdam

## ABSTRACT

Combinatorial optimization lies at the core of many real-world problems. Especially since the rise of graph neural networks (GNNs), the deep learning community has been developing solvers that derive solutions to NP-hard problems by learning the problem-specific solution structure. However, reproducing the results of these publications proves to be difficult. We make three contributions. First, we present an open-source benchmark suite for the NP-hard MAXIMUM INDEPENDENT SET problem, in both its weighted and unweighted variants. The suite offers a unified interface to various state-of-the-art traditional and machine learning-based solvers. Second, using our benchmark suite, we conduct an in-depth analysis of the popular guided tree search algorithm by Li et al. [NeurIPS 2018], testing various configurations on small and large synthetic and real-world graphs. By re-implementing their algorithm with a focus on code quality and extensibility, we show that the graph convolution network used in the tree search does not learn a meaningful representation of the solution structure, and can in fact be replaced by random values. Instead, the tree search relies on algorithmic techniques like graph kernelization to find good solutions. Thus, the results from the original publication are not reproducible. Third, we extend the analysis to compare the tree search implementations to other solvers, showing that the classical algorithmic solvers often are faster, while providing solutions of similar quality. Additionally, we analyze a recent solver based on reinforcement learning and observe that for this solver, the GNN is responsible for the competitive solution quality.

## 1 INTRODUCTION

Various communities have been dealing with the question of how to efficiently solve combinatorial problems, which frequently are NP-hard. These problems often have real-world applications in industry, for instance in staff assignment (Peters et al., 2019), supply chain optimization (Eskandarpour et al., 2015), and traffic optimization (Böther et al., 2021).

In recent years, also the machine learning community has been engaged in solving combinatorial problems. One reason that learning-based approaches are interesting, even though commercial general purpose solvers like Gurobi (Gurobi Optimization LLC, 2021) exist, is that we might be able to learn the solution structure of a specific *family of instances*, e.g., similar vehicle routing problems instances are solved everyday (Khalil et al., 2017; Dong et al., 2021). Such a trained model can be used together with an algorithmic component to find feasible solutions quickly; we outline in Section 2.1 the designs that past works have used.

However, these statistics-based techniques have the well-known problem of reproducibility (Ioannidis, 2005; Baker, 2016), which is a particular problem in machine learning research with untested or even unpublished code and data sets (Kapoor & Narayanan, 2020; Ding et al., 2020), and is the

---

[*]Correspondence to `maxi@boether.de`

reason for the interest in initiatives such as Papers With Code[1] and the ReScience Journal[2]. Keeping the importance of reproducibility in mind, we evaluate various machine learning approaches for combinatorial optimization, and compare them to traditional solvers. We focus on the MAXIMUM (WEIGHTED) INDEPENDENT SET problem (Miller & Muller, 1960; Karp, 1972), as the previous works we analyze have centered around variants of this problem. Overall, we contribute the following.

- We provide an open-source, extensible benchmark suite for MAXIMUM (WEIGHTED) INDEPENDENT SET solvers. The software currently supports five state-of-the-art solvers, INTEL-TREESEARCH (Li, Chen, and Koltun, 2018), GUROBI (Gurobi Optimization LLC, 2021), KAMIS (Lamm et al., 2017; Hespe et al., 2019; Lamm et al., 2019), LEARNING WHAT TO DEFER (Ahn, Seo, and Shin, 2020), and DGL-TREESEARCH. Our DGL-TREESEARCH is a modern re-implementation of the INTEL-TREESEARCH, implemented in PyTorch (Paszke et al., 2019) and the Deep Graph Library (Wang et al., 2019), with a focus on clean, readable code, as well as performance, and it fixes various issues of the original code. Our evaluation suite lays the ground for further research on hard combinatorial (graph) problems and aims at providing a fair and comparable environment for further evaluations.

- Using our re-implementation of the tree search, we propose and analyze additional techniques aiming at improving the guided search. Employing the benchmark suite, we conduct an exhaustive analysis of various configurations of the tree search algorithms, showing that the results of the highly-cited INTEL-TREESEARCH approach are not reproducible, neither with the original code nor with our re-implementation. When exploring the design space further, we show that the various techniques used by the tree search algorithm to improve the results, like graph kernelization, are the reason for good performance, especially on hard data sets. In fact, replacing the GNN output with random values performs similar to using the trained network.

- Having analyzed the configuration space, we compare the tree search approaches to the classical solvers like GUROBI and KAMIS, showing that problem-tailored solvers are often the superior approach. Without using techniques like graph reduction in the tree search, classical solvers are superior. The classical solvers show to be more efficient even when accessing these routines – that have been implemented for the algorithmic solvers in the first place – in the tree search. Last, we show that LEARNING WHAT TO DEFER seems to be able to find good results very quickly, indicating that unsupervised reinforcement learning for combinatorial problems is a promising direction for future research.

## 2 INDEPENDENT SET SOLVERS AND MACHINE LEARNING FOR COMBINATORIAL OPTIMIZATION

In this section, we formally introduce the MAXIMUM INDEPENDENT SET (MIS) problem as well as the solvers included in our analysis, and discuss related work in the broader space of deep learning for combinatorial optimization. Given an undirected graph $G = (V, E)$, an independent set is a set of vertices $S \subseteq V$ for which for all vertices $u, v \in S$, $(u, v) \notin E$. For $u \in V$, let $w_u$ be its weight, and let $\text{IS}(G)$ be all independent sets of $G$, then the MAXIMUM WEIGHTED INDEPENDENT SET (MWIS) problem aims at determining $\arg\max_{S \in \text{IS}(G)} \sum_{u \in S} w_u$. The unweighted MIS problem is equivalent to the MWIS problem, where f.a. $u \in V$, $w_u = 1$. Both problems are strongly NP-complete (Garey & Johnson, 1978). Next, we briefly explain the solvers that we use to find such MIS.

**Gurobi.** GUROBI is a commercial mathematical optimization solver. There are various ways of formulating the M(W)IS problem mathematically (Butenko, 2003). In the main paper, we formulate the MWIS problem as the linear program above, and discuss other variants in Appendix D.

**KaMIS.** KAMIS is an open-source solver tailored towards the MIS and MWIS problems. It offers support both for the unweighted case (Lamm et al., 2017; Hespe et al., 2019) as well as the weighted case (Lamm et al., 2019). It employs graph kernelization and an optimized branch-and-bound algorithm to efficiently find independent sets. Note that the algorithms and techniques differ between the weighted and unweighted cases. We use the code unmodified from the official repository[3].

---

[1] https://paperswithcode.com/
[2] https://rescience.github.io/
[3] https://github.com/KarlsruheMIS/KaMIS

**Intel-TreeSearch.** In their influential paper, Li et al. (2018) propose a guided tree search algorithm to find maximum independent sets of a graph. The idea is to train a graph convolutional network (GCN) (Kipf & Welling, 2017), which assigns each vertex a probability of belonging to the independent set, and then greedily and iteratively assign vertices to the set. They furthermore employ the reduction and local search algorithms by KAMIS to speed up the computation. We use their published code[4], which unfortunately is not runnable in its default state. We apply a git patch[5] to make the code runnable, enable further evaluation by collecting statistics, and add command-line flags for more fine-grained control of the solver configuration. In Appendix C, we give some details on possible complications with the original code. Because a good knowledge of the algorithm is important to follow the remainder of this paper, we briefly describe the algorithm. A pseudocode description of the algorithm is found in Algorithm 1. The core element of the tree search is a queue[6] $P$ of partial solutions $S \in \{0, 1, \perp\}^{|V|}$ to the MIS problem, i.e., labelings of the graph at hand marking each vertex as either *included* in the MIS (1), *excluded* (0), or *unlabeled* (a decision is still to be made, $\perp$). To be a valid (partial) solution to the MIS problem, each element in the queue fulfills the constraint that no two adjacent vertices can both be included. Furthermore, all vertices adjacent to an *included* one must be *excluded*.

Given a graph $G = (V, E)$, we start with an empty solution, i.e., f.a. $v \in V$, $S_v = \perp$. In each step of the tree search, a partial labeling is popped from the queue, and the *residual graph* $G_{\text{residual}}$ consisting only of the unlabeled vertices is constructed. Next, we obtain a predefined number $m$ of *probability maps*, each of which contains a value for all vertices of the residual graph, posing the "probability of being in the MIS", by calling the trained GCN, which outputs its assignments from vertices to probabilities, i.e., $\text{GCN}(G) \in [0, 1]^{|V| \times m}$.

From each of these probability maps, a new partial solution is derived as follows: The probabilities of the maps are sorted in descending order. The vertex with the highest probability gets labeled as *included*, and all adjacent vertices as *excluded*. This step is repeated until we would have to label an already *excluded* vertex as *included*, in which case we break, and add the partial solution to the queue. In case all vertices are labeled we obtained a full solution. This procedure is repeated for all probability maps. For further modifications of this algorithm (e.g., reduction), we refer to Appendix B.

**DGL-TreeSearch.** Because the code provided by Li et al. (2018) might be difficult to read and maintain, and hence is prone to errors in the evaluation, we re-implement the tree search using PyTorch (Paszke et al., 2019) and the established Deep Graph Library (Wang et al., 2019). Our implementation aims at offering a more readable and modern implementation, which benefits from improvements in the two deep learning libraries during recent years. Furthermore, it fixes various issues of the original implementation that sometimes deviates from the paper. Additionally, we implement further techniques to improve the search, like queue pruning, and weighted selection of the next element, as well as multi-GPU functionality.

**Learning What To Defer.** We test LEARNING WHAT TO DEFER (LwD), an unsupervised deep reinforcement learning-based solution introduced by Ahn et al. (2020). Their idea is similar to the tree search, as the algorithm iteratively assigns vertices to the independent set. However, this is not done using a supervised GCN, but instead by an unsupervised agent built upon the GraphSAGE architecture (Hamilton et al., 2017) and trained by Proximal Policy Optimization (Schulman et al., 2017). There is no queue of partial solutions. We refer to the original paper for details on the algorithm. As their code[7] does not work with generic input, we patch their code.

Our open-source benchmarking suite[8] integrates all these solvers in one easily accessible command-line interface using Anaconda (Anaconda Inc., 2020), with a unified input and output format. We provide our code for DGL-TREESEARCH and our GUROBI interface directly, and download, compile, and patch the other solvers on-demand. It handles the correct invocation of the solvers and allows to quickly run experiments on various solvers in different configurations.

---

[4]https://github.com/isl-org/NPHard

[5]A git patch is a file transparently stating the changes we require to make the code compatible with our benchmarking suite.

[6]We use the term *queue* to stay consistent with other works. However, $P$ is not a traditional LIFO queue, but a list (infinite-sized array).

[7]https://github.com/sungsoo-ahn/learning_what_to_defer

[8]Repository: https://github.com/MaxiBoether/mis-benchmark-framework

## 2.1 RELATED WORK

**AI for Combinatorial Optimization.** Research at the intersection of artificial intelligence and combinatorial optimization is not limited to MIS. These methods differ in the problem they tackle, and how algorithmic components and learned components interact. The evolutionary computation community, for example, has researched various NP-hard routing problems, like the VEHICLE ROUTING PROBLEM (Berger & Barkaoui, 2003; Potvin, 2009) and the MULTIPLE ROUTES problem (Böther et al., 2021). In a nutshell, evolutionary optimization is an iterative improvement that is not guided by a learned component, but instead tries to mutate and combine existing solutions randomly, in order to find better solutions. ML-based solvers often are variants of branch-and-bound solvers. In general, such solvers partition the search space by some rule set; for ML-based solvers, the partition rules are not pre-defined, but given by a model that learned how to branch (Balcan et al., 2018). The concrete algorithms differ in the model used, how they utilize the model to branch within the search space, and how the model is trained. For example, the INTEL-TREESEARCH trains a GCN on pre-labeled data, and expects from its guidance model a probability for each vertex that describes how likely it is that the vertex belongs to the MIS, and then greedily operates on these probabilities. In comparison to that, LwD instead models the branching process as an unsupervised agent that can choose vertices in a Markov Decision Process, that similarly to the tree search iteratively picks vertices, but for example allows to roll-back invalid solutions, instead of greedily pushing forward. This methodology applies to other combinatorial problems as well. For example, Kool et al. (2019) propose an attention-based encoder-decoder-architecture as a model for solving TRAVELLING SALESMAN PROBLEM (TSP) instances, where the decoder iteratively outputs probabilties for each vertex to be visited next. They analyze different algorithmic components that utilize that model, for example, greedily following the most probable vertex, or sampling multiple tours and picking the best. Other research has shown that the question of scaling such architectures to real-world instances poses a challenge by itself, and that currently, algorithmic solvers often outperform ML-based solvers for larger instances (Joshi et al., 2021). A theoretical understanding of how such models extrapolate to different instance families is also topic of current research (Xu et al., 2021).

With this discussion, we want to show that the design space for ML-based solvers is broad, as on the one hand, one has to design a model that learns how to solve a problem, and on the other hand build an efficient algorithmic component that utilizes that model to actually find a solution. These components can become very complex, as seen in recent work by Nair et al. (2021) which aims at solving Mixed Integer Programs and uses two neural network components instead of one; the *neural diving* component finds variable assignments, while the *neural branching* component guides the branch-and-bound algorithm in its next step.

**Other Solvers for Maximum Independent Set.** Next to the solvers included in this paper, there are some other solvers available. Khalil et al. (2017) propose S2V-DQN, in which they use Q-learning to solve the minimum vertex cover problem[9]. Compared to LwD, which marks multiple vertices as part of the MIS in a single step, S2V-DQN only labels a single vertex in each step. We mention the recent publication by Hespe et al. (2021) introducing some new reduction rules for MIS, which have not yet been included in KAMIS. Another heuristic for MIS that we do not consider in this paper in favor of KAMIS is GRASP (Feo et al., 1994). Note that all of these solvers are heuristics and hence only solve MIS approximately; exact solvers have been proposed by Jain & Seshadhri (2020); Xiao & Nagamochi (2017); Tomita et al. (2010), for example, and theory has been working on understanding why and in what models MIS poses to be difficult (Censor-Hillel et al., 2017).

## 3 EVALUATION

In this section, we first introduce our experimental setup and then focus on the analysis of the supervised tree search approach for combinatorial optimization in Section 3.1. After having derived a good configuration for the tree search algorithm, we continue to compare the tree search algorithms to the other classical and reinforcement learning solvers in Section 3.2. We investigate the scalability of the approaches in Section 3.3, and analyze the behavior on the weighted MIS problem in Section 3.4.

**Experimental Setup.** We run all our experiments on an NVIDIA DGX-1 with two 20-Core Intel Xeon E5-2698 CPUs at 2.2 GHz, leading to overall 40 physical and 80 logical cores, 512 GB of

---

[9]MVC is very related to the maximum independent set, as one can just flip the assignment.

Table 1: Results of the tree searches in various configurations; the full table with more datasets and more configuration options can be found in Table 2. For all configurations, in the first row, we state the average MIS size as well as the average approximation factor. In the second row, the average time in seconds until the best solution was found, and, in brackets, the number of graphs where any solution was found are given. The average values refer only to the graphs within a dataset for which a solution was found. For Intel, here we show the default setting (`d`) and the setup with both reduction and local search enabled (`r+ls`). For DGL, we explore the configuration space further, as we analyze the default (`d`), reduction and local search (`r+ls`), queue pruning and weighted queue pop (`qp+wp`), and the full configuration where we replace the GCN by randomly generated outputs (`+rand`). For the random graphs of size 50-100, all solvers have a time limit of 15 seconds; for the SATLIB and PPI datasets the time limit is 30 seconds; for VC-BM and DIMACS graphs, the time limit is 5 minutes.

| | | Intel | | DGL | | | |
|---|---|---|---|---|---|---|---|
| Graph | Nodes | d | r+ls | d | r+ls | qp+wp | qp+wp+r+ls+rand |
| ER | 50-100 | 20.58 (0.98) 1.70 (500) | **20.83 (1.00)** 0.02 (500) | 19.88 (0.95) 3.53 (500) | **20.83 (1.00)** 0.28 (500) | 19.16 (0.92) 3.39 (500) | **20.83 (1.00)** 0.31 (500) |
| | 700-800 | 39.90 (-) 12.00 (100) | **44.08 (-)** 16.81 (100) | 37.13 (-) 13.63 (100) | 43.90 (-) 12.82 (100) | 34.79 (-) 9.11 (100) | 44.02 (-) 10.08 (100) |
| HRG | 50-100 | 33.62 (0.99) 3.24 (495) | **33.72 (1.00)** 0.00 (500) | 32.80 (0.97) 5.50 (500) | **33.72 (1.00)** 0.06 (500) | 30.93 (0.92) 3.45 (500) | **33.72 (1.00)** 0.07 (500) |
| | 700-800 | - (-) - | **304.21 (1.00)** 0.03 (100) | 221.80 (0.75) 17.81 (5) | **304.21 (1.00)** 0.40 (100) | 245.29 (0.80) 13.55 (98) | **304.21 (1.00)** 0.44 (100) |
| SATLIB | 1209-1347 | - (-) - | 426.39 (0.99) 6.89 (500) | 340.50 (0.79) 18.48 (2) | 426.25 (0.99) 9.55 (500) | 356.93 (0.84) 17.91 (211) | **426.48 (0.99)** 5.58 (500) |
| VC-BM | 450-1534 | 39.85 (-) 149.80 (40) | 44.26 (-) 166.04 (39) | 37.20 (-) 139.55 (40) | 44.35 (-) 84.08 (40) | 35.42 (-) 68.33 (40) | **44.58 (-)** 64.57 (40) |
| DIMACS | 125-4000 | 37.14 (-) 79.30 (37) | 76.41 (-) 43.12 (37) | 57.68 (-) 87.53 (37) | 76.49 (-) 36.36 (37) | 45.54 (-) 42.56 (37) | **76.51 (-)** 27.89 (37) |
| PPI | 591-3480 | - (-) - | **1002.83 (1.00)** 24.70 (24) | 269.00 (0.97) 23.56 (1) | 1002.67 (0.99) 3.70 (24) | 804.38 (0.92) 20.11 (13) | 1002.79 (0.99) 4.11 (24) |

memory, and a total of eight NVIDIA Tesla V100 GPUs. For each experiment, we explicitly state the number of threads and GPUs used. We run Ubuntu 20.04.1 LTS, using Linux kernel `4.15.0-124`.

**Datasets.** We evaluate the various solvers and configurations using both real-world datasets as well as generated random graphs in various sizes. We make use of the random graph models by Erdős & Rényi (1960) (ER), Albert & Barabási (2002) (BA), Holme & Kim (2002) (HK), Watts & Strogatz (1998) (WS) and the Hyperbolic Random Graph model (Krioukov et al., 2010) (HRG). The graph generation functionality, employing NetworkX (Hagberg et al., 2008) and girgs (Bläsius et al., 2019) as backends, is integrated into our benchmark suite. For real-world datasets, we focus on the hard SATLIB dataset (Hoos & Stützle, 2000), which consists of synthetic 3-SAT instances, the vertex cover benchmark (VC-BM) (Xu et al., 2007) consisting of 40 graphs on which finding the maximum independent set is synthetically made hard, and the DIMACS challenge graphs; we also test various other graphs, like citation networks. Details on all mentioned datasets as well as hyperparameters of graph generation can be found in Appendix E.

### 3.1 ANALYSIS OF THE TREE SEARCH

In this subsection, we analyze the INTEL-TREESEARCH and DGL-TREESEARCH. These supervised approaches require training a GCN (Kipf & Welling, 2017). Following Li et al. (2018), we train both on the SATLIB dataset for 20 epochs using the Adam optimizer (Kingma & Ba, 2015). As the GCN outputs multiple probability maps (c.f. Appendix B), we employ the hindsight loss, which, for multiple choices, outputs the loss of the best choice (Guzmán-Rivera et al., 2012; Chen & Koltun, 2017). We fix the number of probability maps to 32, to enable comparison with Li et al. (2018). We test the solvers in different configurations on various graphs; these configurations are assorted with increasing levels of complexity and intuitively should improve the solution quality. Details on what effects the individual configuration options have are outlined in Appendix B. An excerpt of our results can be found in Table 1; the full table can be found in the Appendix A in Table 2.

**Random Graph Results.** First, we discuss the results of the random graphs. For all small graphs, both tree searches in all configurations find solutions most of the time. These are close to optimal as soon as reduction (`+r`) and local search (`+ls`) are involved. Notably, on larger non-ER random graphs, in

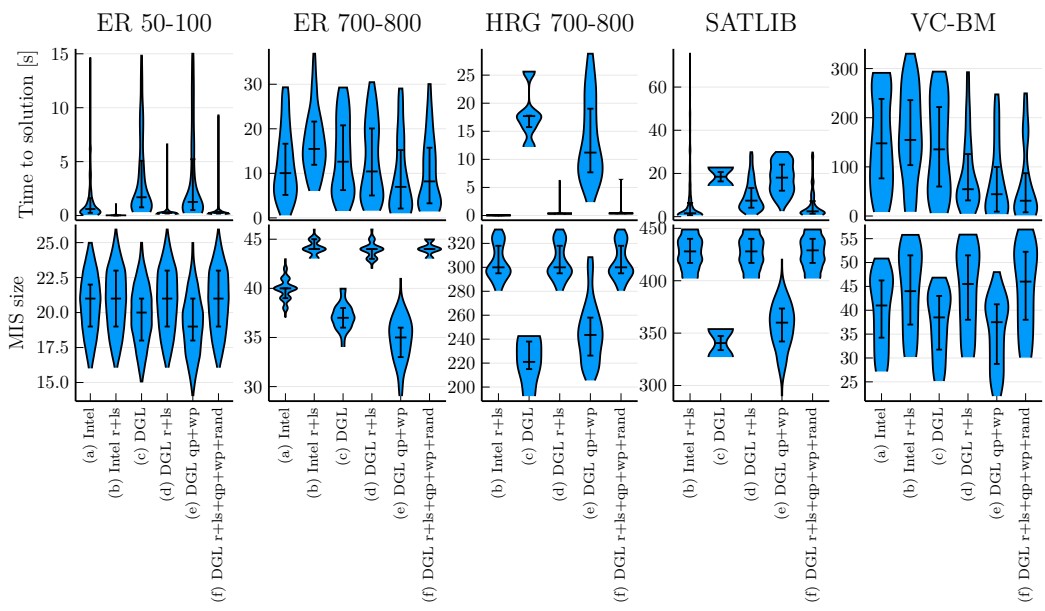

Figure 1: Violin plots of time to solution and MIS sizes of the tree searches in various configurations on a selection of data sets.

the default variants, where neither reduction nor local search is enabled, both INTEL-TREESEARCH and DGL-TREESEARCH do not often find a solution. This shows that for medium-sized graphs, vanilla tree searches cannot discover solutions within a feasible time limit. The reductions often single-handedly solve the problem instance, as seen for example on the simple BA graphs, where a solution is found instantaneously.

**Queue Pruning and Weighted Queue Pop.** To approach the issue of requiring hand-tailored reduction techniques to obtain any solution, with DGL-TREESEARCH, we analyze queue pruning and the weighted queue pop. While queue pruning itself performs similar to the default configuration for all graphs, the weighted pop vastly increases the number of solutions found. For example, for large hyperbolic random graphs, in the default configuration, the DGL-TREESEARCH was only able to find solutions for 5 graphs (and the INTEL-TREESEARCH was not able to find any solution), whereas using queue pruning together with the weighted queue pop, we find solutions for 98 % of all large HRGs. A similar observation can be made for WS graphs. Interestingly, it seems like the MIS problem is harder to address with the default approach on graphs that try to model real-world networks, such as the just mentioned HRGs and WS graphs (Bläsius et al., 2018). Overall, queue pruning might be desirable to reduce memory consumption, but its impact on solving quality and time is limited; on the other hand, weighted queue popping is a general idea that brings some more depth-search-like behavior into the breadth-search-like approach at hand, and thus enables the solver to find a lot more solutions.

**Importance of Reduction.** For small and large BA, HK, WS graphs as well as HRGs, the reduction itself already leads to an achieved average approximation of 1. Only for the large ER graphs, for which GUROBI was not able to find provably optimal assignments, the local search further improves the average independent set size. Multithreading cannot improve the results of the random graphs.

**Hard Real-World Graph Results.** Now, we discuss the results for the hard benchmark datasets. Unlike previously, where we just aimed at finding an MIS on random graphs, solving the MIS problem on SATLIB is equivalent to finding a satisfiable assignment to synthesized hard 3-SAT instances. Thus, one can expect this data set to be particularly hard. Our experiments confirm this, as both implementations rarely find solutions within the time limit without reduction enabled. We can see that, similar to the random graphs, the weighted queue pop enables the search to at least find *some* solution, as this configuration is at least able to find 211 instead of no results at all. However, the

average independent set size is 357, which is not very good, considering each MIS in this dataset contains at least 403 vertices.

Considering VC-BM, due to the higher time limit, the default configurations are already able to find some solution. Interestingly, for VC-BM, the reductions do not improve the performance of the solvers. Hence, the weighted queue pop is very important for finding results quickly, shrinking the time needed from the default 140 seconds to under 70 seconds. For DIMACS, we see similar behavior to SATLIB, i.e., the reduction and local search techniques improve the average MIS, and the weighted queue pop enables faster discovery of solutions.

**Impact of the Findings.** These results are interesting for various reasons. First, the models are trained on SATLIB instances, but the default configuration does not find any independent set for other instances of the this family, which was one motivation for using machine learning for optimization (c.f. Section 1). The results contradict the original paper by Li et al. (2018), which claims an average MIS of 426 and related work by Ahn et al. (2020), claiming an average MIS of 418. Additionally to our own trained weights, we test the model weights provided in the official INTEL-TREESEARCH repository, but do not find any differences in the results. Ahn et al. (2020) mention that they modified the official INTEL-TREESEARCH code. Unfortunately, neither the paper documents how exactly their queue pruning has been implemented, nor were the authors themselves able to provide us their modifications to the original INTEL-TREESEARCH repository when we contacted them. Hence, it remains unclear whether the difference in the SATLIB results between Ahn et al. (2020) and our experiments stems from how they implemented queue pruning, or from some unwanted side effects in their experiments (e.g., the reduction may have been still enabled). Furthermore, we unsuccessfully contacted Li et al. (2018) about our findings. As both the INTEL-TREESEARCH *and* our re-implementation DGL-TREESEARCH, which was written from scratch, exhibit this behavior, we suspect that there must be an undocumented modification or problem in the experiments that leads to Ahn et al. (2020) obtaining rather good results with the default configuration. Overall, neither the original code with the original weights, nor the original code with newly trained weights, nor our reimplementation are performing as originally claimed. In order to be as transparent as possible about our findings, we provide all of our code changes applied to the Intel repository as a patch and provide the entire source code of the re-implemented DGL-TREESEARCH as well.

**Replacing the GCN with Random Values.** Next, we analyze the replacement of the outputs of the GCN with random values. We start with the randomized default configuration, i.e., no techniques like reduction are enabled. In this case, we find that for all small random graphs, the tree search is still able to find solutions; however, the quality of these MIS seems to be slightly worse than the default results. For 80 % of the larger random graphs – except for ER graphs – the randomized tree search is not able to find solutions. As the default configuration is not able to find any solution for SATLIB, it is to be expected that the random configuration does not find any solution either.

**Randomness for Other Real-World Graphs.** For the other real-world graphs, the reddit datasets are the only datasets where the default configuration performed reasonably well. We find that the randomness here in fact increases the performance of the algorithm.

**Robustness.** Most ouf our datasets consist of multiple instances. In order to shed light onto the distribution of both the solution size as well as the time until the solution was found, in Figure 1, we show violin plots for some data sets and tree search configurations. Note in most real world scenarios, we do not know when we can terminate, hence the run time will always be the maximum time limit configured. In these plots, we can for example again confirm that the combination of queue pruning and weighted popping enables us to find solutions quicker. We also see that as soon as reduction and local search are enabled, the solution distributions of the tree searches are very similar, no matter whether whether we query the GCN or use random values; for SATLIB, we see that using random values the distribution of the time to solution is even narrower towards 0, because no GPU computation is required.

**Final Considerations.** We conclude that the guidance by the GCN does not help our tree search algorithm; for some datasets, it is not able to generalize well (randomness beats the GCN), for others, there is no performance difference between random values and the GCN. If we focus on the configuration of the tree search that could be considered the *production* version[10], i.e., with at least

---

[10]Recall that the default version is not able to solve the SATLIB dataset without the help of reduction and local search techniques.

reduction and local search enabled, we find that random outputs lead to identical performance, with very minor differences in the time until the final solution discovery. Overall, major contributing factors to solution quality are the reduction and local search by KAMIS, and in fact, the tree search algorithm is a "smarter brute-force" approach that does not gain any performance by being guided by a machine learning model. Instead, the search space is only narrowed by techniques such as data reduction and weighted queue popping, instead of good guidance by the ML model.

## 3.2 COMPARING TREE SEARCH SOLVERS TO OTHER SOLVERS

Having understood the implications of the various possible configurations of tree searches, next, we compare INTEL-TREESEARCH and DGL-TREESEARCH to KAMIS, a heuristic solver optimized for the MAXIMUM INDEPENDENT SET problem, the mathematical optimization tool GUROBI, and the reinforcement learning tool LEARNING WHAT TO DEFER. For LwD, we train for 20 000 iterations of proximal policy optimization on the SATLIB dataset, using the hyperparameters given for SATLIB in Table 5 (Appendix A.1) of Ahn et al. (2020). Due to space constraints, the detailed results are given in Appendix A in Table 3.

**KAMIS and GUROBI.** As we can see, the sophisticated state-of-the-art solver KAMIS can solve almost all instances perfectly; for example, for the large ER graphs, the multithreaded INTEL-TREESEARCH has a slightly higher average MIS, and a faster time until a solution is found. For other graphs, especially the SATLIB dataset, KAMIS is very fast, while obtaining very high-quality results. These results are in line with Ahn et al. (2020), who also observed KAMIS outperforming the tree search. Regarding the general-purpose solver GUROBI, we find that it performs similar to KAMIS on simple instances, however, on harder datasets such as SATLIB or VC-BM, we see that Gurobi takes significantly more time to find good solutions, and the average MIS is a little smaller (e.g., for large ER graphs, KAMIS achieves 44.57, compared to 37.79 for GUROBI). Note that GUROBI and KAMIS have to rely on the CPU, while the tree search employs one one or more GPUs, unless it uses random values. Evening when assigning eight V100 GPUs to the multithreaded tree search, the computational advantage of the tree search does not help to outperform the other solvers, showing that the workload is not GPU-bound.

**LwD.** We analyze LwD first in its default configuration and then, similar to the tree search, replace the output of the graph neural network with a random tensor. First, LwD is very fast, even though it requires neural network inference. For example, on the DIMACS data set, LwD finds solutions on average in just 4 seconds, while KAMIS takes 121 seconds. Quality-wise, except for the VC-BM dataset, it always finds near-optimal solutions. Note that we did not use the local search or reduction techniques for LwD, which could further improve solution quality. When using random output instead of the neural network, unlike the tree search algorithms, the solution quality degrades noticeably. These promising results show that LwD did not only learn the solution structure of the SATLIB instance family, but additionally is able to generalize over different datasets.

**Robustness.** Due to space constraints, we show violin plots of the results in Appendix A in Figure 2. We clearly see the impact of randomness on LwD, and can visually compare the time difference between GUROBI and KAMIS, for example on VC-BM.

**Final Considerations.** For tree search approaches, we see that state-of-the-art algorithmic techniques are required to find good solutions. As the fully-configured versions of the tree searches employ these KAMIS-internal routines, one can argue that the purely algorithmic solvers are the better choice, because the quality difference is negligible, while KAMIS is faster than the tree searches. Purely algorithmic solvers do not come with the overhead of a machine learning environment (training as well as execution are more complicated), and the important algorithmic techniques need to be developed in any case. Our results are in line with observations by Joshi et al. (2021) who have compared algorithmic and classical solvers to deep learning solutions for TSP, and observe that DL-based solutions are often outperformed, especially on larger instances.

## 3.3 LARGE-SCALE GRAPHS

However, the question is whether these observations also hold true for large-scale graphs. To this end, we evaluate the solvers on random graph instances from 500 000 to 5 000 000 vertices, and on huge real-world graphs. Detailed results can be found in Appendix A in Table 4.

Overall, we observe similar performance characteristics on large-scale graphs. Only KAMIS and GUROBI are able to find solutions for all scenarios. KAMIS performs best, both with respect to solution quality and time required to find these solutions; in many cases, it is more than one order of magnitude faster than other solvers. Compared to the INTEL-TREESEARCH, the DGL-TREESEARCH time outs more often due to VRAM limits; the INTEL-TREESEARCH benefits from lower-level implementation using numpy arrays and sparse adjacency matrices, while the DGL-TREESEARCH employs the higher level Deep Graph Library graph abstraction, consuming more memory. For LwD, we see good results for graphs up to 500 000 vertices, but cannot verify the scalability up to 2 000 000 vertices of the original paper[11].

## 3.4 WEIGHTED GRAPHS

Having analyzed the solvers on unweighted graphs, we briefly want to see how the solvers perform on weighted graphs. Intuitively, the weighted problem is harder, because each vertex can have a different value; hence specialized reduction techniques have to be developed. Lamm et al. (2019) were the first to present such rules also for the weighted case that are implemented in KAMIS, however, only integer weights are currently supported. We test weighted random graphs and three Amazon MWIS data sets. The results and details on the weighted graph generation are given in Appendix A in Table 5; note that INTEL-TREESEARCH does not support the weighted case, and while Ahn et al. (2020) evaluate LwD for MWIS, they do not provide the code to do so.

We observe that on HK, WS, and HRG graphs, where KAMIS is able to utilize its reduction techniques, it is able to instantaneously solve the weighted problem as well. However, on ER graphs, the reduction times out, showing that the suitability of the algorithmic techniques is graph-dependent. In the weighted case on ER graphs, the DGL-TREESEARCH with queue pruning enabled finds the best result. GUROBI finds solutions of similar quality to KAMIS very quickly. For the Amazon data sets, all solvers except GUROBI reach their limits. KAMIS does not support the large node weights that are used in the Amazon instances and hence is not able to solve them. The DGL-TREESEARCH goes out of memory and crashes while solving these instances. Overall, we see that both GUROBI as well as the vanilla tree search have the advantage of being problem-agnostic, i.e., an extension to the weighted case was easily possible, while for the algorithmic solver, new reductions are needed, that currently cannot deal with some graphs. On smaller random graphs where the reductions are successful, KAMIS finds solutions of the highest quality, showing the trade-off between general and specialized solvers. GUROBI is the only solver that is able to solve large, weighted, real-world MWIS instances, and shows that industry-standard optimizers are very robust towards different inputs of different sizes, while smaller algorithmic solvers need more time to reach that level of robustness.

## 4 CONCLUSION AND FUTURE WORK

We present our comprehensive, open-source benchmark environment for the MAXIMUM (WEIGHTED) INDEPENDENT SET problem. Using this environment, we run several experiments on both real-world and synthetic graphs of different sizes. Our analysis shows that guided tree searches, such as the INTEL-TREESEARCH by Li et al. (2018), owe their good results not to the trained neural network, but instead to the various techniques used to make a "better" brute-force algorithm. To verify this, we show that the GCN that guides the search can be replaced by random values without a noticeable performance impact. Furthermore, we are not able to reproduce the results of previous work in the default configuration of the algorithm and claim that without algorithmic techniques, the tree search algorithms are not able to solve hard MIS instances. We believe our results to be an important insight for the community researching at the intersection of combinatorial optimization and machine learning. The benchmark suite lays the ground fur future reproducible evaluations for new MIS solvers, and the promising results for LEARNING WHAT TO DEFER indicate that reinforcement learning is superior to supervised approaches. This might be kept in mind when developing solving techniques using machine learning in the future. For future work, further variants of the MAXIMUM INDEPENDENT SET problem, such as the GENERALIZED INDEPENDENT SET problem (Colombi et al., 2017; Hosseinian & Butenko, 2019), and other problems, such as TSP, should be considered, to further understand for what kind of problem what solver architecture should be used.

---

[11] As Ahn et al. (2020) do not explicitly state the hyperparameter configuration for their large-scale experiments and random graph generation, these experiments might not be directly comparible

ETHICS STATEMENT

We adhere to the ICLR Code of Ethics and declare that we have no conflicts of interest. We tried to contact the authors of the related work, as stated in the respective sections.

REPRODUCIBILITY STATEMENT

As our results contradict previous work, we try to be as transparent as possible about our process. The solvers we use are documented in Section 2. The changes we apply to LwD and INTEL-TREESEARCH are supplied as a git patch with our benchmark suite that can be accessed via the our repository[12] or in the supplementary material published alongside with this paper. The datasets we use are listed in Appendix E; we supply the code required to preprocess the datasets.

ACKNOWLEDGMENTS

We thank the anonymous reviewers for their invaluable feedback and the HPI FutureSOC Lab for computing infrastructure.

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

## A  ADDITIONAL TABLES AND PLOTS

In this section, you can find the tables and plots omitted in the main paper due to space constraints.

Table 2: Results of the tree searches in various configurations. We run experiments on both synthetic and real-world graphs with varying numbers of nodes (c.f. Appendix E). For all configurations, in the first row, we state the average MIS size as well as the average approximation factor for graphs where we were able to pre-calculate the provably optimal MIS using GUROBI. In the second row, the average time in seconds until the best solution was found, and, in brackets, the number of graphs where any solution was found are given. The average values refer only to the graphs within a dataset for which a solution was found. Columns that start with a plus (+) add an additional flag to the previous column; columns that do not start with a plus use *just* the stated flags. For Intel, we start with the default setting (d), proceed to include the graph reduction (+r), and then the local search (+ls), and last test reduction and local search in a multithreaded setup (+mt). For DGL, we explore the configuration space further. After analyzing the default (d), adding reduction (+r), and local search (+ls), we test configurations replacing the GCN with random outputs (rand), using queue pruning (qp), and weighted queue pop (+wp). We also test a combination of reduction, queue pruning, and weighted queue pop (+r), proceed to add the local search (+ls), and again replace the GCN by randomly generated outputs (+rand). The full configuration is tested with and without random output in a multithreaded setup (+mt). All multithreaded experiments in this table use 8 threads, and all threads share a single GPU. For the random graphs of size 50-100, all solvers have a time limit of 15 seconds; for the SATLIB, as-caida, PPI, REDDIT, Citeseer, Cora, and ego-facebook datasets the time limit is 30 seconds; for the bitcoin, PubMed, Wikipedia, VC-BM, DIMACS, and roadnet-berlin graphs, the time limit is 5 minutes. We refer to Appendix B for detailed explanations of the techniques used. We mark the configuration/solver that has the best average MIS in bold, and grey out configurations for which solutions were found for less than 20 % of all graphs.

| Graph | Nodes | Intel d | +r | +ls | +mt | DGL d | +r | +ls | rand | qp | +wp | +r | +ls | +rand | r+qp+wp+ls+mt | +rand |
|---|---|---|---|---|---|---|---|---|---|---|---|---|---|---|---|---|
| ER | 50-100 | 20.58 (0.98) 1.70 (500) | 20.76 (0.99) 0.81 (500) | 20.83 (1.00) 0.02 (500) | 20.83 (1.00) 5.39 (500) | 19.88 (0.95) 3.53 (500) | 20.58 (0.98) 1.89 (500) | 20.83 (1.00) 0.28 (500) | 19.05 (0.91) 5.26 (500) | 19.69 (0.94) 2.66 (500) | 19.16 (0.92) 3.39 (500) | 20.54 (0.98) 1.59 (500) | 20.83 (1.00) 0.26 (500) | 20.83 (1.00) 0.31 (500) | 20.83 (0.99) 0.41 (500) | 20.82 (0.99) 0.37 (499) |
| ER | 700-800 | 39.90 (-) 12.00 (100) | 39.71 (-) 19.73 (98) | 44.08 (-) 16.81 (100) | 44.98 (-) 11.90 (100) | 37.13 (-) 13.63 (100) | 38.51 (-) 15.91 (100) | 43.90 (-) 12.82 (100) | 15.71 (100) | 14.76 (100) | 9.11 (100) | 12.52 (100) | 9.30 (100) | 10.08 (100) | 44.32 (-) 22.97 (96) | 44.55 (-) 17.95 (99) |
| BA | 50-100 | 42.42 (0.99) 0.54 (500) | 42.43 (1.00) 0.00 (500) | 42.43 (1.00) 0.00 (500) | 42.43 (1.00) 4.97 (500) | 42.05 (0.99) 2.44 (500) | 42.43 (1.00) 0.08 (500) | 42.43 (1.00) 0.50 (500) | 40.81 (0.96) 5.52 (500) | 41.72 (0.98) 1.84 (500) | 41.74 (0.98) 1.64 (500) | 42.43 (1.00) 0.07 (500) | 42.43 (1.00) 0.08 (500) | 42.43 (1.00) 0.11 (500) | 42.43 (1.00) 0.12 (500) | 42.43 (1.00) 0.12 (500) |
| BA | 700-800 | 420.89 (0.97) 3.09 (18) | 432.58 (1.00) 0.01 (100) | 432.58 (1.00) 0.02 (100) | 432.58 (1.00) 5.31 (100) | 389.65 (0.95) 16.50 (80) | 432.58 (1.00) 0.43 (100) | 432.58 (1.00) 0.45 (100) | 374.37 (0.86) 20.08 (19) | 398.90 (0.92) 19.86 (3) | 406.82 (0.94) 6.92 (100) | 432.58 (1.00) 0.43 (100) | 432.58 (1.00) 0.67 (100) | 432.58 (1.00) 0.69 (100) | 432.58 (1.00) 0.73 (100) | 432.58 (1.00) |
| HK | 50-100 | 42.32 (0.99) 0.53 (500) | 42.33 (1.00) 0.00 (500) | 42.33 (1.00) 0.00 (500) | 42.33 (1.00) 0.00 (500) | 41.87 (0.98) 2.59 (500) | 42.33 (1.00) 0.06 (500) | 42.33 (1.00) 0.07 (500) | 40.87 (0.96) 5.78 (500) | 41.50 (0.98) 2.08 (500) | 41.45 (0.98) 1.77 (500) | 42.33 (1.00) 0.07 (500) | 42.33 (1.00) 0.07 (500) | 42.33 (1.00) 0.07 (500) | 42.33 (1.00) 0.07 (500) | 42.33 (1.00) 0.08 (500) |
| HK | 700-800 | 417.89 (0.98) 4.23 (19) | 429.81 (1.00) 0.01 (100) | 429.81 (1.00) 0.01 (100) | 429.81 (1.00) 5.26 (100) | 390.64 (0.90) 19.66 (83) | 429.81 (1.00) 0.37 (100) | 429.81 (1.00) 0.38 (100) | 370.05 (0.86) 22.29 (19) | 393.10 (0.91) 18.15 (80) | 407.04 (0.94) 7.50 (100) | 429.81 (1.00) 0.42 (100) | 429.81 (1.00) 0.39 (100) | 429.81 (1.00) 0.44 (100) | 429.81 (1.00) 0.42 (100) | 429.81 (1.00) 0.43 (100) |
| WS | 50-100 | 38.69 (0.99) 0.53 (500) | 38.70 (1.00) 0.00 (500) | 38.70 (1.00) 0.00 (500) | 38.70 (1.00) 0.00 (500) | 37.91 (0.97) 3.80 (500) | 38.70 (1.00) 0.07 (500) | 38.70 (1.00) 0.07 (500) | 36.58 (0.94) 5.22 (500) | 37.60 (0.97) 2.42 (500) | 36.94 (0.95) 2.17 (500) | 38.70 (1.00) 0.08 (500) | 38.70 (1.00) 0.08 (500) | 38.70 (1.00) 0.08 (500) | 38.70 (1.00) 0.07 (500) | 38.70 (1.00) 0.08 (500) |
| WS | 700-800 | - (-) | 386.90 (1.00) 0.00 (100) | 386.90 (1.00) 0.00 (100) | 386.90 (1.00) 5.49 (100) | - | 386.90 (1.00) 0.37 (100) | 386.90 (1.00) 0.36 (100) | 326.71 (0.87) 21.34 (7) | - | 368.64 (0.95) 13.02 (87) | 386.90 (1.00) 0.40 (100) | 386.90 (1.00) 0.42 (100) | 386.90 (1.00) 0.43 (100) | 386.90 (1.00) 0.39 (100) | 386.90 (1.00) 0.43 (100) |
| HRG | 50-100 | 33.62 (0.99) 3.24 (495) | 33.72 (1.00) 0.00 (500) | 33.72 (1.00) 0.00 (500) | 33.72 (1.00) 0.00 (500) | 32.80 (0.97) 5.50 (500) | 33.72 (1.00) 0.07 (500) | 33.72 (1.00) 0.07 (500) | 32.40 (0.96) 5.14 (500) | 32.62 (0.96) 4.23 (500) | 30.93 (0.92) 3.45 (500) | 33.72 (1.00) 0.06 (500) | 33.72 (1.00) 0.07 (500) | 33.72 (1.00) 0.07 (500) | 33.72 (1.00) 0.07 (500) | 33.72 (1.00) 0.07 (500) |
| HRG | 700-800 | - | 304.21 (1.00) 0.01 (100) | 304.21 (1.00) 0.01 (100) | 304.21 (1.00) 5.51 (100) | 221.80 (0.75) 17.81 (5) | 304.21 (1.00) 0.35 (100) | 304.21 (1.00) 0.40 (100) | 260.27 (0.86) 20.57 (15) | 221.67 (0.74) 21.23 (6) | 245.29 (0.80) 13.55 (98) | 304.21 (1.00) 0.39 (100) | 304.21 (1.00) 0.39 (100) | 304.21 (1.00) 0.44 (100) | 304.21 (1.00) 0.42 (100) | 304.21 (1.00) |
| SATLIB | 1209-1347 | - (-) 11.54 (500) | 424.86 (0.99) 6.89 (500) | 426.39 (0.99) 7.99 (500) | 426.51 (1.00) | 340.50 (0.79) 18.48 (2) | 421.93 (0.98) 16.42 (500) | 426.25 (0.99) 9.55 (500) | - (-) | - (-) | 356.93 (0.84) 17.91 (211) | 419.61 (0.98) 6.23 (500) | 426.37 (0.99) 5.58 (500) | 426.48 (0.99) 13.12 (500) | 426.55 (0.99) 10.34 (500) | 426.57 (0.99) |
| VC-BM | 450-1534 | 39.85 (-) 149.80 (40) | 39.08 (-) 169.49 (38) | 44.26 (-) 166.04 (39) | 45.10 (-) 51.64 (40) | 37.20 (-) 139.55 (40) | 38.70 (-) 138.83 (40) | 44.35 (-) 84.08 (40) | 35.45 (-) 84.08 (40) | 37.70 (-) 119.78 (40) | 35.42 (-) 68.33 (40) | 38.38 (-) 93.88 (40) | 44.52 (-) 73.67 (40) | 44.58 (-) 64.57 (40) | 44.65 (-) 145.24 (40) | 44.88 (-) 113.43 (40) |
| DIMACS | 125-4000 | 37.14 (-) 79.30 (37) | 74.56 (-) 90.29 (36) | 76.41 (-) 43.12 (37) | 76.86 (-) 21.26 (37) | 57.68 (-) 87.53 (37) | 72.76 (-) 88.80 (37) | 76.49 (-) 36.36 (37) | 30.06 (-) 105.90 (34) | 60.97 (-) 60.59 (37) | 45.54 (-) 42.56 (37) | 71.70 (-) 31.51 (37) | 76.57 (-) 34.11 (37) | 76.51 (-) 27.89 (37) | 79.58 (-) 45.64 (26) | 79.27 (-) 20.61 (26) |
| as-Caida | 8020-26475 | - (-) | 19387.47 (0.99) 0.26 (122) | 19387.47 (0.99) 8.41 (122) | 19387.47 (0.99) | - (-) | 19388.03 (1.00) 7.32 (122) | 19387.90 (0.99) 11.22 (122) | - (-) | - (-) | - (-) | 19388.03 (1.00) 7.62 (122) | 19387.90 (0.99) 12.27 (122) | 19387.90 (0.99) 14.48 (122) | 19387.90 (0.99) 15.63 (122) | 19387.90 (0.99) 12.74 (122) |
| PPI | 591-3480 | - (-) 14.59 (24) | 1002.12 (0.99) 14.59 (24) | 1002.83 (1.00) 24.70 (24) | 1002.71 (0.99) 8.73 (24) | 269.00 (0.97) 23.56 (1) | 1001.83 (0.99) 7.14 (24) | 1002.67 (0.99) | - (-) | 372.67 (0.84) 19.86 (3) | 804.38 (0.92) 20.11 (13) | 1001.67 (0.99) 9.81 (24) | 1002.67 (0.99) 3.68 (24) | 1002.79 (0.99) 4.11 (24) | 1002.50 (0.99) 3.70 (24) | 1002.58 (0.99) 4.82 (24) |
| REDDIT-B | 46-2283 | 146.06 (0.99) 5.94 (200) | 287.54 (1.00) 0.00 (500) | 287.54 (1.00) 0.02 (500) | 287.54 (1.00) 5.37 (500) | 186.31 (0.87) 11.81 (406) | 287.54 (1.00) 0.18 (500) | 287.54 (1.00) 0.20 (500) | 248.96 (0.96) 15.23 (475) | 198.68 (0.89) 12.29 (424) | 249.27 (0.92) 4.93 (492) | 287.54 (1.00) 0.20 (500) | 287.54 (1.00) 0.24 (500) | 287.54 (1.00) 0.26 (500) | 287.54 (1.00) 0.26 (500) | 287.54 (1.00) 0.27 (500) |
| REDDIT-5K | 55-3648 | 194.27 (0.99) 7.97 (37) | 586.31 (0.99) 0.01 (500) | 586.31 (1.00) 0.04 (500) | 586.31 (1.00) 5.35 (500) | 252.74 (0.90) 15.36 (152) | 586.32 (1.00) 0.33 (500) | 586.32 (1.00) 0.37 (500) | 378.83 (0.94) 19.41 (325) | 255.66 (0.89) 17.87 (168) | 416.49 (0.92) 9.63 (396) | 586.32 (1.00) 0.37 (500) | 586.32 (1.00) 0.46 (500) | 586.32 (1.00) 0.44 (500) | 586.32 (1.00) 0.48 (500) | 586.32 (1.00) 0.50 (500) |
| REDDIT-12K | 41-897 | 128.61 (0.99) 3.99 (349) | 170.20 (1.00) 0.00 (500) | 170.20 (1.00) 0.01 (500) | 170.20 (1.00) 5.20 (500) | 136.05 (0.92) 9.53 (467) | 170.20 (1.00) 0.12 (500) | 170.20 (1.00) 0.14 (500) | 163.15 (0.98) 11.46 (497) | 145.23 (0.95) 8.17 (478) | 155.81 (0.93) 3.86 (500) | 170.20 (1.00) 0.14 (500) | 170.20 (1.00) 0.16 (500) | 170.20 (1.00) 0.16 (500) | 170.20 (1.00) 0.17 (500) | 170.20 (1.00) 0.17 (500) |
| wiki-RfA | 11380 | - (-) | 8111.00 (1.00) 0.86 | 8111.00 (1.00) 6.50 | 8111.00 (1.00) 13.08 | - (-) | 8096.00 (0.99) 10.79 | 8096.00 (0.99) 16.14 | - (-) | - (-) | 8072.00 (0.99) 270.94 | 8096.00 (0.99) 12.41 | 8096.00 (0.99) 19.12 | 8096.00 (0.99) 20.40 | 8096.00 (0.99) 19.95 | 8096.00 (0.99) 26.38 |
| wiki-Vote | 7115 | - (-) | 4866.00 (1.00) 0.45 | 4866.00 (1.00) 2.60 | 4866.00 (1.00) 8.56 | - (-) | 4864.00 (0.99) 7.79 | 4864.00 (0.99) 12.05 | - (-) | - (-) | 4748.00 (0.97) 236.68 | 4864.00 (0.99) 10.82 | 4864.00 (0.99) 13.99 | 4864.00 (0.99) 19.11 | 4864.00 (0.99) 11.20 | 4864.00 (0.99) 24.41 |
| PubMed | 19717 | - (-) | 15912.00 (1.00) 0.25 | 15912.00 (1.00) 6.07 | 15912.00 (1.00) 12.91 | - (-) | 15895.00 (0.99) 12.32 | 15888.00 (0.99) 17.12 | - (-) | - (-) | - (-) | 15888.00 (0.99) 14.77 | 15888.00 (0.99) 18.44 | 15888.00 (0.99) 20.44 | 15888.00 (0.99) 23.34 | 15888.00 (0.99) 29.13 |
| Cora | 2708 | - (-) | 1451.00 (1.00) 0.03 | 1451.00 (1.00) 0.21 | 1451.00 (1.00) 5.63 | - (-) | 1451.00 (1.00) 7.88 | 1451.00 (1.00) 6.27 | - (-) | - (-) | - (-) | 1451.00 (1.00) 9.73 | 1451.00 (1.00) 11.48 | 1451.00 (1.00) 21.09 | 1451.00 (1.00) 13.63 | 1451.00 (1.00) 6.38 |
| Citeseer | 3264 | - (-) | 1808.00 (1.00) 0.03 | 1808.00 (1.00) 0.24 | 1808.00 (1.00) 5.51 | - (-) | 1808.00 (1.00) 9.21 | 1808.00 (1.00) 6.19 | - (-) | - (-) | - (-) | 1808.00 (1.00) 9.60 | 1808.00 (1.00) 11.96 | 1808.00 (1.00) 17.24 | 1808.00 (1.00) 14.62 | 1808.00 (1.00) 7.34 |
| bitcoin-alpha | 3783 | - (-) | 2718.00 (1.00) 0.05 | 2718.00 (1.00) 0.50 | 2718.00 (1.00) 6.10 | - (-) | 2716.00 (0.99) 6.65 | 2716.00 (0.99) 5.28 | - (-) | - (-) | 2658.00 (0.97) 76.54 | 2716.00 (0.99) 8.58 | 2716.00 (0.99) 8.91 | 2716.00 (0.99) 12.33 | 2716.00 (0.99) 11.41 | 2716.00 (0.99) 16.35 |
| bitcoin-otc | 5881 | - (-) | 4346.00 (1.00) 0.08 | 4346.00 (1.00) 1.13 | 4346.00 (0.99) 6.86 | - (-) | 4352.00 (1.00) 7.44 | 4352.00 (1.00) 6.19 | - (-) | - (-) | 4352.00 (1.00) 10.33 | 4352.00 (1.00) 10.35 | 4352.00 (1.00) 12.95 | 4352.00 (1.00) | 4352.00 (1.00) 12.47 | 4352.00 (1.00) 14.57 |
| ego-facebook | 4039 | - (-) | - (-) | 1046.00 (1.00) 14.34 | 1046.00 (1.00) 14.34 | 1030.00 (0.98) 17.04 | 1046.00 (1.00) 11.82 | 1046.00 (1.00) | - (-) | - (-) | 1030.00 (0.98) 29.87 | 1046.00 (1.00) 14.94 | 1046.00 (1.00) 23.95 | 1046.00 (1.00) | 1046.00 (1.00) 14.12 | 1046.00 (1.00) 14.12 |
| roadnet-berlin | 61204 | - (-) | - (-) | - (-) | 29843.00 (0.99) 144.22 | - (-) | 29792.00 (0.99) 28.27 | 29792.00 (0.99) 34.13 | - (-) | - (-) | - (-) | 29792.00 (0.99) 33.04 | 29792.00 (0.99) 39.53 | 29792.00 (0.99) 39.21 | 29888.00 (1.00) 68.22 | 29888.00 (1.00) 52.70 |

Table 3: Results of the modern MIS solver KAMIS, the optimization tool GUROBI, the reinforcement-learning based LEARNING WHAT TO DEFER, and the tree search algorithms in their default and full configuration. For the explanation of the various configuration flags, we refer to the caption of Table 2. All multithreaded experiments in this table use 8 threads, and all threads share a single GPU. Time limits are set equally to Table 2. We note that KAMIS and LwD always run single-threadedly, while GUROBI employs up to 8 threads, as needed. For LwD, we test the default configuration (d) and replace the output of the GNN with a random tensor (rand); the random graphs are executed with 32 iterations per episode, and all other graphs with 128 iterations per episode (c.f. Ahn et al. (2020)). We mark the configuration/solver that has the best average MIS in bold, and grey out configurations for which solutions were found for less than 20 % of all graphs.

| Graph | Nodes | Intel d | Intel r+ls+mt | DGL d | DGL r+qp+wp+ls+mt | LwD d | LwD +rand | KaMIS default | Gurobi default |
|---|---|---|---|---|---|---|---|---|---|
| ER | 50-100 | 20.58 (0.98) | **20.83 (1.00)** | 19.88 (0.95) | **20.83 (0.99)** | 20.36 (0.97) | 8.64 (0.41) | **20.83 (1.00)** | **20.83 (1.00)** |
| | | 1.70 (500) | 5.39 (500) | 3.53 (500) | 0.41 (500) | 0.24 (500) | 0.06 (500) | 1.60 (500) | 0.10 (500) |
| | 700-800 | 39.90 (-) | **44.98 (-)** | 37.13 (-) | 44.32 (-) | 33.65 (-) | 8.85 (-) | 44.57 (-) | 37.79 (-) |
| | | 12.00 (100) | 11.90 (100) | 13.63 (100) | 22.97 (96) | 0.37 (100) | 0.12 (100) | 31.28 (100) | 30.01 (100) |
| BA | 50-100 | 42.42 (0.99) | **42.43 (1.00)** | 42.05 (0.99) | **42.43 (1.00)** | **42.43 (1.00)** | 22.38 (0.53) | **42.43 (1.00)** | **42.43 (1.00)** |
| | | 0.54 (500) | 4.97 (500) | 2.44 (500) | 0.11 (500) | 0.26 (500) | 0.06 (500) | 0.05 (500) | 0.00 (500) |
| | 700-800 | *420.89 (0.97)* | **432.58 (1.00)** | 389.65 (0.90) | **432.58 (1.00)** | 432.57 (0.99) | 187.79 (0.43) | **432.58 (1.00)** | **432.58 (1.00)** |
| | | *3.09 (18)* | 5.31 (100) | 16.50 (80) | 0.69 (100) | 0.28 (100) | 0.10 (100) | 0.06 (100) | 0.01 (100) |
| HK | 50-100 | 42.32 (0.99) | **42.33 (1.00)** | 41.87 (0.98) | **42.33 (1.00)** | **42.33 (1.00)** | 22.40 (0.53) | **42.33 (1.00)** | **42.33 (1.00)** |
| | | 0.53 (500) | 5.15 (500) | 2.59 (500) | 0.07 (500) | 0.25 (500) | 0.06 (500) | 0.06 (500) | 0.00 (500) |
| | 700-800 | *417.89 (0.98)* | **429.81 (1.00)** | 390.64 (0.90) | **429.81 (1.00)** | 429.77 (0.99) | 188.94 (0.43) | **429.81 (1.00)** | **429.81 (1.00)** |
| | | *4.23 (19)* | 5.26 (100) | 19.66 (83) | 0.42 (100) | 0.09 (100) | 0.09 (100) | 0.08 (100) | 0.01 (100) |
| WS | 50-100 | 38.69 (0.99) | **38.70 (1.00)** | 37.91 (0.97) | **38.70 (1.00)** | 38.70 (0.99) | 24.59 (0.63) | **38.70 (1.00)** | **38.70 (1.00)** |
| | | 0.53 (500) | 5.24 (500) | 3.80 (500) | 0.07 (500) | 0.24 (500) | 0.07 (500) | 0.04 (500) | 0.00 (500) |
| | 700-800 | - (-) | **386.90 (1.00)** | - (-) | **386.90 (1.00)** | **386.90 (1.00)** | 220.07 (0.56) | **386.90 (1.00)** | **386.90 (1.00)** |
| | | - | 5.49 (100) | - | 0.39 (100) | 0.28 (100) | 0.09 (100) | 0.05 (100) | 0.00 (100) |
| HRG | 50-100 | 33.62 (0.99) | **33.72 (1.00)** | 32.80 (0.97) | **33.72 (1.00)** | **33.72 (1.00)** | 19.51 (0.58) | **33.72 (1.00)** | **33.72 (1.00)** |
| | | 3.24 (495) | 5.11 (500) | 5.50 (500) | 0.06 (500) | 0.26 (500) | 0.07 (500) | 0.06 (500) | 0.00 (500) |
| | 700-800 | - (-) | **304.21 (1.00)** | *221.80 (0.75)* | **304.21 (1.00)** | 304.07 (0.99) | 140.20 (0.46) | **304.21 (1.00)** | **304.21 (1.00)** |
| | | - | 5.51 (100) | *17.81 (5)* | 0.44 (100) | 0.30 (100) | 0.11 (100) | 0.13 (100) | 0.01 (100) |
| SATLIB | 1209-1347 | - (-) | 426.51 (0.99) | *340.50 (0.79)* | 426.55 (0.99) | 423.48 (0.99) | 144.85 (0.33) | **426.59 (0.99)** | 426.57 (0.99) |
| | | - | 7.99 (500) | *18.48 (2)* | 13.12 (500) | 0.91 (500) | 0.09 (500) | 4.51 (500) | 3.12 (500) |
| VC-BM | 450-1534 | 39.85 (-) | **45.10 (-)** | 37.20 (-) | 44.65 (-) | 36.02 (-) | 9.78 (-) | 44.95 (-) | 42.73 (-) |
| | | 149.80 (40) | 51.64 (40) | 139.55 (40) | 145.24 (40) | 1.95 (40) | 0.15 (40) | 86.58 (40) | 266.71 (40) |
| DIMACS | 125-4000 | 37.14 (-) | 76.86 (-) | 57.68 (-) | **79.58 (-)** | 70.11 (-) | 35.84 (-) | 76.78 (-) | 74.35 (-) |
| | | 79.30 (37) | 21.26 (37) | 87.53 (37) | 45.64 (26) | 4.13 (37) | 0.26 (37) | 120.88 (37) | 202.18 (37) |
| as-Caida | 8020-26475 | - (-) | 19387.47 (0.99) | - (-) | **19387.90 (1.00)** | 19387.47 (0.99) | 7267.39 (0.37) | 19387.47 (0.99) | 19387.47 (0.99) |
| | | - | 14.49 (122) | - | 15.63 (122) | 3.65 (122) | 6.22 (122) | 2.23 (122) | 0.15 (122) |
| PPI | 591-3480 | - (-) | 1002.71 (0.99) | *269.00 (0.97)* | 1002.50 (0.99) | 1001.54 (0.99) | 251.62 (0.25) | **1002.83 (1.00)** | **1002.83 (1.00)** |
| | | - | 8.73 (24) | *23.56 (1)* | 3.70 (24) | 1.67 (24) | 0.28 (24) | 5.86 (24) | 8.16 (24) |
| REDDIT-B | 46-2283 | 146.06 (0.99) | **287.54 (1.00)** | 186.31 (0.87) | **287.54 (1.00)** | **287.54 (1.00)** | 147.79 (0.52) | **287.54 (1.00)** | **287.54 (1.00)** |
| | | 5.94 (200) | 5.37 (500) | 11.81 (406) | 0.26 (500) | 0.90 (500) | 0.08 (500) | 0.18 (500) | 0.00 (500) |
| REDDIT-5K | 55-3648 | *194.27 (0.99)* | 586.31 (0.99) | 252.74 (0.88) | **586.32 (1.00)** | 586.31 (0.99) | 287.78 (0.49) | 586.31 (0.99) | 586.31 (0.99) |
| | | *7.97 (37)* | 5.35 (500) | 15.36 (152) | 0.48 (500) | 0.93 (500) | 0.10 (500) | 0.22 (500) | 0.00 (500) |
| REDDIT-12K | 41-897 | 128.61 (0.99) | **170.20 (1.00)** | 136.05 (0.92) | **170.20 (1.00)** | **170.20 (1.00)** | 91.45 (0.55) | **170.20 (1.00)** | **170.20 (1.00)** |
| | | 3.99 (349) | 5.20 (500) | 9.53 (467) | 0.17 (500) | 0.86 (500) | 0.07 (500) | 0.10 (500) | 0.00 (500) |
| wiki-RfA | 11380 | - (-) | **8111.00 (1.00)** | - (-) | 8096.00 (0.99) | 8107.00 (0.99) | 2243.00 (0.27) | **8111.00 (1.00)** | **8111.00 (1.00)** |
| | | - | 13.08 | - | 19.95 | 4.79 | 3.07 | 0.82 | 1.57 |
| wiki-Vote | 7115 | - (-) | **4866.00 (1.00)** | - (-) | 4864.00 (0.99) | 4864.00 (0.99) | 1430.00 (0.29) | **4866.00 (1.00)** | **4866.00 (1.00)** |
| | | - | 8.56 | - | 11.20 | 3.94 | 2.18 | 0.73 | 0.87 |
| PubMed | 19717 | - (-) | **15912.00 (1.00)** | - (-) | 15888.00 (0.99) | **15912.00 (1.00)** | 5474.00 (0.34) | **15912.00 (1.00)** | **15912.00 (1.00)** |
| | | - | 12.91 | - | 23.34 | 3.34 | 2.02 | 0.16 | 0.19 |
| Cora | 2708 | - (-) | **1451.00 (1.00)** | - (-) | **1451.00 (1.00)** | **1451.00 (1.00)** | 644.00 (0.44) | **1451.00 (1.00)** | **1451.00 (1.00)** |
| | | - | 5.63 | - | 13.63 | 2.74 | 1.25 | 0.06 | 0.03 |
| Citeseer | 3264 | - (-) | **1808.00 (1.00)** | - (-) | **1808.00 (1.00)** | **1808.00 (1.00)** | 928.00 (0.51) | **1808.00 (1.00)** | **1808.00 (1.00)** |
| | | - | 5.51 | - | 14.62 | 3.38 | 1.74 | 0.06 | 0.04 |
| bitcoin-alpha | 3783 | - (-) | **2718.00 (1.00)** | - (-) | 2716.00 (0.99) | **2718.00 (1.00)** | 1011.00 (0.37) | **2718.00 (1.00)** | **2718.00 (1.00)** |
| | | - | 6.10 | - | 11.41 | 2.37 | 1.85 | 0.21 | 0.05 |
| bitcoin-otc | 5881 | - (-) | 4346.00 (1.00) | - (-) | **4352.00 (1.00)** | 4346.00 (0.99) | 1574.00 (0.36) | 4346.00 (0.99) | 4346.00 (0.99) |
| | | - | 6.86 | - | 12.47 | 2.71 | 2.21 | 0.06 | 0.08 |
| ego-facebook | 4039 | - (-) | **1046.00 (1.00)** | - (-) | - (-) | 1034.00 (0.98) | 291.00 (0.27) | **1046.00 (1.00)** | **1046.00 (1.00)** |
| | | - | 14.34 | - | - | 5.50 | 1.84 | 19.86 | 30.05 |
| roadnet-berlin | 61204 | - (-) | 29843.00 (0.99) | - (-) | **29888.00 (1.00)** | 29739.00 (0.99) | 14520.00 (0.49) | 29843.00 (0.99) | 29843.00 (0.99) |
| | | - | 144.22 | - | 68.22 | 6.66 | 2.17 | 1.95 | 22.72 |

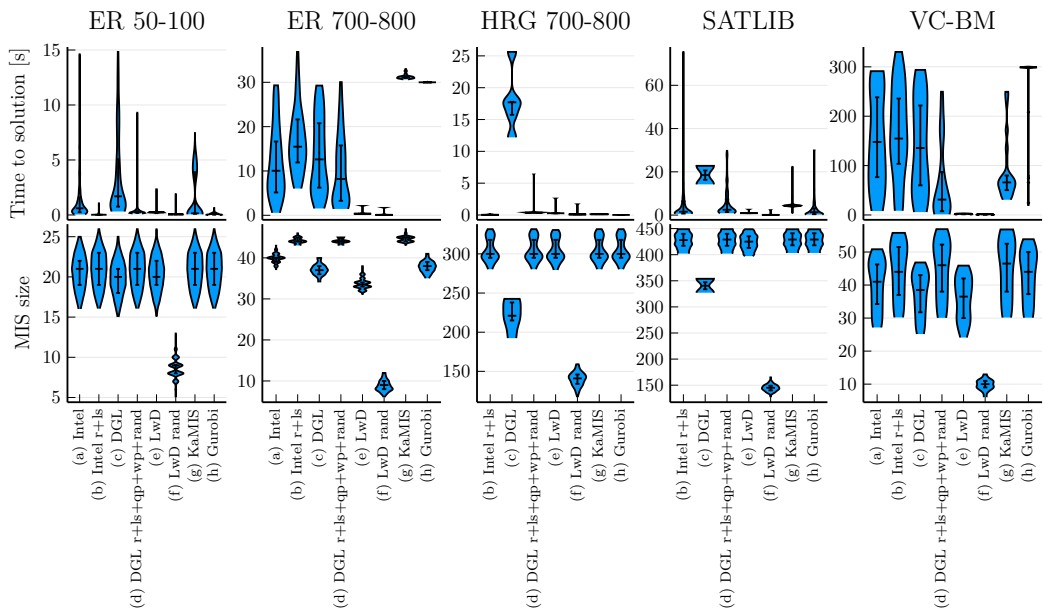

Figure 2: Violin plots of time to solution and MIS sizes of the modern MIS solver KAMIS, the optimization tool GUROBI, the reinforcement-learning based LEARNING WHAT TO DEFER, and the tree search algorithms in their default and full configuration on a selection of data sets.

Table 4: Results of the modern MIS solver KAMIS, the optimization tool GUROBI, the reinforcement-learning based LEARNING WHAT TO DEFER, and the tree search algorithms in their full configuration, on huge random graphs and huge real world networks. We do not test huge ER graphs due to the time complexity of generating them, and do not test BA graphs as they are similar to HK graphs. For the explanation of the various configuration flags, we refer to the caption of Table 2. All multithreaded experiments in this table use 8 threads, and all threads share a single GPU. The time limit for all graphs was set to 3 hours. For LwD, the algorithm is executed with with 128 iterations per episode (c.f. Ahn et al. (2020)). We mark the configuration that has the best average MIS in bold. Configurations that consumed too much VRAM or DRAM are noted as out of memory (OOM).

| | | Intel | DGL | LwD | KaMIS | Gurobi |
|---|---|---|---|---|---|---|
| Graph | Nodes | r+ls+mt | r+qp+wp+ls+mt | default | default | default |
| HK | 500 000 | **289486.00 (-)** 242.91 | **289486.00 (-)** 177.12 | **289486.00 (-)** 145.40 | **289486.00 (-)** 0.27 | **289486.00 (-)** 26.86 |
| | 2 000 000 | **1157643.00 (-)** 1064.53 | OOM | OOM | **1157643.00 (-)** 1.15 | **1157643.00 (-)** 154.14 |
| | 5 000 000 | **2892242.00 (-)** 1207.98 | OOM | OOM | **2892242.00 (-)** 3.80 | **2892242.00 (-)** 446.81 |
| WS | 500 000 | **258887.00 (-)** 219.57 | **258887.00 (-)** 171.97 | **258887.00 (-)** 61.28 | **258887.00 (-)** 0.22 | **258887.00 (-)** 5.41 |
| | 2 000 000 | **1035715.00 (-)** 923.82 | **1035715.00 (-)** 740.14 | OOM | **1035715.00 (-)** 0.95 | **1035715.00 (-)** 28.48 |
| | 5 000 000 | **2589519.00 (-)** 1128.76 | OOM | OOM | **2589519.00 (-)** 2.82 | **2589519.00 (-)** 76.26 |
| HRG | 500 000 | **210950.00 (-)** 374.53 | **210950.00 (-)** 216.09 | 210930.00 (-) 834.99 | **210950.00 (-)** 0.78 | 210949.00 (-) 47.20 |
| | 2 000 000 | **843618.00 (-)** 1114.92 | OOM | OOM | **843618.00 (-)** 3.51 | 843616.00 (-) 374.15 |
| | 5 000 000 | **2110401.00 (-)** 1866.42 | OOM | OOM | **2110401.00 (-)** 8.62 | 2110400.00 (-) 897.20 |
| DBLP | 317 080 | **152131.00 (1.00)** 142.15 | **152131.00 (1.00)** 112.32 | **152131.00 (1.00)** 236.98 | **152131.00 (1.00)** 0.30 | **152131.00 (1.00)** 7.15 |
| roadnet-pennsylvania | 1 088 092 | OOM | OOM | OOM | 533625.00 (-) 17.49 | **533628.00 (-)** 3662.87 |
| ego-gplus | 107 614 | **57394.00 (-)** 399.20 | OOM | 56794.00 (-) 275.58 | **57394.00 (-)** 115.76 | 57321.00 (-) 10802.52 |
| web-google | 875 713 | **529138.00 (1.00)** 663.18 | OOM | 528725.00 (0.99) 822.14 | **529138.00 (1.00)** 10.67 | **529138.00 (1.00)** 83.30 |

Table 5: Results of the MIS solver KAMIS, the optimization tool GUROBI, and the DGL-TREESEARCH in various configurations on weighted random graphs, and Amazon MWIS instances. The weights on the random garphs have been sampled per vertex from a normal distribution $\mathcal{N}$ with $\mu = 100, \sigma = 30$, capped to be larger than 0, and rounded to integer values. For the explanation of the various configuration flags, we refer to the caption of Table 2. All multithreaded experiments in this table use 8 threads, and all threads share a single GPU. The time limit for all graphs was set to 30 seconds. We mark the configuration that has the best average MIS in bold, and grey out configurations for which solutions were found for less than 20 % of all graphs. Similar to the unweighted experiments, we run the solvers on 500 graphs per type. Configurations where there was a timeout due to KAMIS' reduction techniques are marked as TO. Configurations where the solver went out of memory are marked as OOM. Note that KAMIS did not support the Amazon instances, as the node weights are too large.

| Graph | Nodes | DGL | | | | | | | | KaMIS | Gurobi |
|---|---|---|---|---|---|---|---|---|---|---|---|
| | | d | +r | +ls | qp | +wp | +r | +ls | +mt | default | default |
| ER | 700-800 | 3765.08 (-) 18.80 (500) | TO | TO | **3792.71 (-)** 18.01 (500) | 3571.05 (-) 11.65 (500) | TO | TO | TO | TO | 3709.68 (-) 30.01 (500) |
| HK | 700-800 | 39469.48 (0.87) 20.11 (364) | 42026.61 (0.93) 1.02 (500) | 45034.05 (0.99) 0.68 (500) | 39128.12 (0.87) 20.18 (329) | 40441.76 (0.89) 11.89 (500) | 42026.61 (0.93) 0.97 (500) | 45034.01 (0.99) 0.75 (500) | 45033.46 (0.99) 0.92 (500) | **45036.91 (0.99)** 0.00 (500) | **45036.91 (0.99)** 0.01 (500) |
| WS | 700-800 | - (-) - | 36010.38 (0.87) 3.13 (500) | 41160.71 (0.99) 5.38 (500) | 37958.50 (0.92) 14.77 (2) | 36904.74 (0.89) 15.17 (439) | 36010.51 (0.87) 3.41 (500) | 41159.49 (0.99) 6.13 (500) | 41147.84 (0.99) 10.19 (500) | **41196.43 (1.00)** 0.00 (500) | **41196.43 (1.00)** 0.01 (500) |
| HRG | 700-800 | 23641.88 (0.70) 23.25 (24) | 33366.88 (0.99) 0.62 (500) | 33696.93 (0.99) 0.57 (500) | 22583.27 (0.68) 24.89 (22) | 24497.41 (0.72) 17.64 (482) | 33366.88 (0.99) 0.60 (500) | 33696.93 (0.99) 0.54 (500) | 33696.93 (0.99) 0.57 (500) | **33696.96 (1.00)** 0.00 (500) | **33696.96 (1.00)** 0.01 (500) |
| AMAZON-MR | 14058-30467 | TO | TO | TO | TO | TO | **inf (-)** | TO | TO | TO | 2147287259.20 (-) |
| AMAZON-MT | 979-12320 | TO | TO | TO | TO | TO | TO | TO | TO | TO | **283567086.67 (-)** |
| AMAZON-MW | 3079-47504 | TO | TO | TO | TO | TO | TO | TO | TO | TO | **706635840.75 (-)** |

# B   TREE SEARCH

In the following, we describe some additional details on the guided tree search approach introduced by Li et al. (2018). A textual description can be found in Section 2, and a pseudocode description in Algorithm 1; for a visualization, we refer to Figure 1 of Li et al. (2018). The pseudocode makes use of a function gen_probmaps : $\mathfrak{G} \to [0,1]^{|V| \times m}$, that can take arbitrary unweighted graphs $G \in \mathfrak{G}$ as input. In the default case, this function calls the trained GCN, which outputs its assignments from vertices to probabilities, i.e., $\mathsf{GCN}(G) \in [0,1]^{|V| \times m}$. We obtain the final solution in a non-pseudocode implementation – like DGL-TREESEARCH – for example by choosing the largest independent set that is yielded.

Furthermore, there are several modifications (flags) discussed in Section 3.1. In the following, we briefly discuss how these flags impact the tree search algorithm.

- **Reduction.** Reduction or graph kernelization is an addition that uses algorithmic techniques to find vertices which (provably) *have to* be in the independent set. After determining the residual graph, before generating the probability maps, we optionally can determine whether some vertices can be labeled due to the reduction rules, and then continue to determine the probability maps with an even smaller residual graph. Lamm et al. (2017) and Hespe et al. (2019) provide reduction rules for the unweighted case, which both INTEL-TREESEARCH and DGL-TREESEARCH access using a Python-C++ interface. Furthermore, for the MAXIMUM WEIGHTED INDEPENDENT SET problem, DGL-TREESEARCH uses the recently proposed reduction rules by Lamm et al. (2019).

- **Local Search.** A local search is an addition based on small mutations which try to further improve a full solution before it is yielded. It can be understood as fine-tuning a solution. The INTEL-TREESEARCH and DGL-TREESEARCH use the local search implementation by Lamm et al. (2017) and Hespe et al. (2019), and for the weighted case, DGL-TREESEARCH uses the local search by Lamm et al. (2019).

- **Multithreading.** To find the MIS faster, one can use multiple threads that search for independent sets. The parallelization idea is very simple, each thread just follows the same procedure as stated in Algorithm 1. In the INTEL-TREESEARCH, the threads share the queue $P$, making push and pop operations a critical section. Because we want to avoid this critical section, and due to issues with shared state in Python multiprocessing, in the

**Algorithm 1:** Guided Treesearch Algorithm to find MIS (adjusted from Li et al. (2018)). For a solution $S$ and vertex $u \in V$, we denote by $S_u$ the current status of $u$ (included, excluded, unlabeled). For a more detailed description, we refer to Appendix B.

**Input:** Graph $G = (V, E)$, Function `gen_probmaps` $: \mathfrak{G} \to [0,1]^{|V| \times m}$.
**Output:** Independent sets $S_i \subseteq V$.

1   $P \leftarrow \{(\perp, \perp, \ldots, \perp) \in \{0, 1, \perp\}^{|V|}\}$;
2   **while** $P$ *is not empty* **do**
3     $S \leftarrow$ random_pop$(P)$;
4     $V_{\text{residual}} \leftarrow \{u \in V \mid S_u = \perp\}$;
5     $G_{\text{residual}} \leftarrow (V_{\text{residual}}, \{(u, v) \in E \mid u, v \in V_{\text{residual}}\})$;
6     **for** $M \in$ gen_probmaps$(G_{\text{residual}})$ **do**
7       $S' \leftarrow S$;
8       **for** $u \in V_{\text{residual}}$ *sorted by descending probability in* $M$ **do**
9         **if** $S'_u = 0$ **then**
10           break;
11         $S'_u \leftarrow 1$;
12         **for** $v \in V_{\text{residual}}$ *adjacent to* $u$ **do**
13           $S'_v \leftarrow 0$;
14       **if** $\forall u \in V : S'_u \neq \perp$ **then**
15         yield $S'$;
16       **else**
17         append $S'$ to $P$;

DGL-TREESEARCH, we give each thread its own queue, and first initialize enough partial solutions, such that each thread has one partial solution to start on.

- **Queue Pruning.** As we see in Section 3.1, on difficult datasets like SATLIB, the tree search almost cannot find any solution. Furthermore, in their experiments, Ahn et al. (2020) state that the INTEL-TREESEARCH runs out of memory, something that we did not observe for medium-sized graphs. Queue pruning is a technique used to tackle these two issues. Unfortunately, we could not obtain the information on how the queue pruning approach by Ahn et al. (2020) is implemented via private communication, hence we propose our own solution. We set a maximum number of elements on our queue $P$, and after appending a new partial solution to the queue, we remove the oldest elements (at position 0) from the queue, until the queue is small enough again. The intuition is that we find more solutions faster, because there is a higher chance to pop an almost labeled (more recent) solution from the queue, and we also limit the memory used. Queue pruning was not proposed in the original paper by Li et al. (2018).

- **Weighted Queue Pop.** By default, the random_pop used in Algorithm 1 chooses an element from the queue uniformly at random. Weighted Queue Pop, on the other hand, shifts the probability of each element to be chosen according to how many unlabeled vertices are left in it, favoring fewer unlabeled vertices. Similar to queue pruning, the intuition is to find fully labeled solutions faster while keeping some randomness in the popping behavior. It can be freely combined with queue pruning or used on its own. To the best of our knowledge, no other paper has proposed this addition to the tree search algorithm yet.

- **Random Values as Probability Maps.** This flag changes gen_probmaps to a random generator, effectively replacing the GCN output with probability values chosen uniformly at random.

We note that the original implementation by Li et al. (2018) does not exactly follow our description and Algorithm 1. We note differences between their paper and the published code in the next section.

## C    REMARKS ON THE INTEL-TREESEARCH IMPLEMENTATION

In this section, we note some important facts to keep in mind when dealing with the original tree search implementation, next to the rather difficult understandability and maintainability of the code. First, for difficult instances, with reduction and local search both disabled, the search queue appears to grow indefinitely without ever finding a single solution. To circumvent this effect and to be able to find solutions, it might be necessary to limit the search queue to a fixed maximum size, as observed by Ahn et al. (2020). They furthermore stress the importance of pruning to limit memory usage. We discuss our approach to queue pruning in Appendix B.

Second, in the paper, it is stated that the GCN is trained for 200 epochs. However, in the implementation, one epoch iterates over just 2000 randomly chosen samples of the 38000 samples in the training set[13]. This is not what you would expect given the term "epoch". Furthermore, the original code has some hardcoded values (e.g., sometimes it expects the number of input graphs to be flexible, and sometimes it is hardcoded within the same file). Our patch provided in the benchmark suite fixes these issues.

Third, the multi-threaded variant of the algorithm (demo_parallel.py) uses just one randomly chosen probability map of the GCN outputs instead of all. This makes this variant's performance in terms of search steps per second, appear to scale superlinearly, and furthermore is not documented in the paper[14], and needs to be kept in mind for evaluation.

## D    DETAILS ON USING GUROBI AS A M(W)IS SOLVER

Gurobi is a general purpose, off-the-shelf, mathematical optimization tool. Hence, many parameters can be tuned, and the performance can very on how the problem to be solved is inputted into the solver. In the following two subsections, we investigate the impact of these two dimensions.

### D.1    GUROBI TUNING

Similar to training machine learning models, we can try to optimize Gurobi's performance for the problem at hand. This is called *hyperparameter tuning* or *algorithm configuration*, and often done using heuristic searches, e.g., by evolutionary algorithms. One well known optimizer for this is SMAC (Lindauer & Hutter, 2018); however, in this subsection, we use grbtune[15], as it is the native tool supported by Gurobi for parameter tuning.

Our methodology is as follows. We generate a Gurobi configuration based on 10 instances of the VC-BM dataset. We choose this data set because for most other datasets like SATLIB, grbtune states that it is not able to improve over the default parameters. We note that grbtune decides when it is done, and we do not impose a time limit on the tuning process. The resulting found parameters are Heuristics = 0.001, PumpPasses = 0, and ZeroObjNodes = 20000000. We then run Gurobi both with the default configuration and the tuned configuration on various data sets, and split the VC-BM data set into the training and test sets.

The results can be seen in Table 6. We find that tuning does not improve the results, and even on the training set leads to worse performance, although the optimization was performed on exactly those instances. Only for the DIMACS data set, the tuned Gurobi finds the solution faster than the default Gurobi configuration. As Gurobi states[16], "The bottom line is that automated performance tuning is meant to give suggestions for parameters that could produce consistent, reliable improvements on your models. It is not meant to be a replacement for efficient modeling or careful performance testing." One possible explanation might be that grbtune might be more effective on more complex linear programs. Additionally, on most data sets, it even states it cannot improve the default parameters. Our

---

[13]https://github.com/isl-org/NPHard/blob/5fc770ce1b1daee3cc9b318046f2361
611894c27/train.py#L92

[14]Algorithm 3 in Appendix C.1 of the paper contains an unbound variable $m$ responsible for this ambiguity.

[15]https://www.gurobi.com/documentation/9.1/refman/command_line_tuning.ht
ml

[16]https://www.gurobi.com/documentation/9.1/refman/parameter_tuning_tool.
html

Table 6: Run times results of the the optimization tool GUROBI with default and tuned parameters (c.f. Appendix D.1) For the explanation of the various configuration flags, we refer to the caption of Table 2. Time limits are set equally to Table 2. GUROBI employs up to 8 threads, as needed. We mark the configuration/solver that has the best average MIS in bold.

| | | Gurobi | |
| Graph | Nodes | default | tuned |
|---|---|---|---|
| ER | 700-800 | **37.79 (-)** 
 30.01 (100) | 36.61 (-) 
 30.02 (100) |
| HRG | 700-800 | **304.21 (1.00)** 
 0.01 (100) | **304.21 (1.00)** 
 0.01 (100) |
| SATLIB | 1209-1347 | 426.57 (0.99) 
 3.12 (500) | **426.58 (0.99)** 
 3.19 (500) |
| VC-BM (Train) | 450-1534 | **32.00 (-)** 
 179.55 (10) | 31.80 (-) 
 205.31 (10) |
| VC-BM (Test) | 450-1534 | **46.30 (-)** 
 300.04 (30) | 45.63 (-) 
 300.07 (30) |
| DIMACS | 125-4000 | **74.35 (-)** 
 202.18 (37) | 73.59 (-) 
 196.87 (37) |
| PPI | 591-3480 | **1002.83 (1.00)** 
 8.16 (24) | 1002.79 (0.99) 
 8.96 (24) |

experiments show that at least for the linear MIS program, the default configuration of a commercial solver like Gurobi are sufficient. This may differ for other problems and hyperparameter tuning should always be considered, but we do not find any benefit on our data sets.

## D.2   ALTERNATIVE MIS FORMULATION

The configuration of the solver is one way we can impact performance. Another way is reformulating the mathematical optimization task into something that the solver can solver more efficiently. Until now, we have used a linear formulation of the MWIS problem, i.e., $\arg\max_{S \in \text{IS}(G)} \sum_{u \in S} w_u$. Literature has discussed different formulations of the MIS problem (Butenko, 2003). We want to see whether using a different program makes a difference on our benchmark datasets. To this end, we use a quadratic formulation of the MIS problem (Pardalos & Rodgers, 1992). Following the notation of Butenko (2003), let $G$ be a graph with $n$ vertices, $A_G$ be its adjacency matrix, and $J$ the $n \times n$ identity matrix. Let $A = J - A_G$.

Then, we can calculate the MIS as

$$\arg\max_{\mathbf{x} \in \{0,1\}^n} \mathbf{x}^T A \mathbf{x},$$

which is a quadratic program instead of a linear one, that however might be easier to solve in practice.

To understand the impact of this formulation, we run Gurobi with this quadratic program instead of the linear one again on our data sets. The results can be seen in Table 7. We find that, with the exception of the DIMACS data set, the linear formulation always performs equally or better, with both respect to solution size and run time. On DIMACS, the quadratic formulation performs maginally better (74.35 average MIS vs 74.62 average MIS), but also requires more time (202 seconds vs 242 seconds). On all other graphs, especially the hard benchmark instances, the linear formulation is faster, for example for VC-BM, where the linear program requires 269 seconds, and the quadratic program 300 seconds.

Table 7: Results of the modern MIS solver KAMIS, the reinforcement-learning based LEARNING WHAT TO DEFER, the tree search algorithms in their default and full configuration, in comparison to GUROBI with the MIS as a linear program (`default`) and quadratic program (`quadratic`). For the explanation of the various configuration flags, we refer to the caption of Table 2. All multithreaded experiments in this table use 8 threads, and all threads share a single GPU. Time limits are set equally to Table 2. We note that KAMIS and LwD always run single-threadedly, while GUROBI employs up to 8 threads, as needed. For LwD, the random graphs are executed with 32 iterations per episode, and all other graphs with 128 iterations per episode (c.f. Ahn et al. (2020)). We mark the configuration/solver that has the best average MIS in bold, and grey out configurations for which solutions were found for less than 20 % of all graphs.

| | | Intel | | DGL | | LwD | KaMIS | Gurobi | |
|---|---|---|---|---|---|---|---|---|---|
| Graph | Nodes | d | r+ls+mt | d | r+qp+wp+ls+mt | d | default | default | quadratic |
| ER | 50-100 | 20.58 (0.98) 1.70 (500) | **20.83 (1.00)** 5.39 (500) | 19.88 (0.95) 3.53 (500) | **20.83 (0.99)** 0.41 (500) | 20.36 (0.97) 0.24 (500) | **20.83 (1.00)** 1.60 (500) | **20.83 (1.00)** 0.10 (500) | **20.83 (1.00)** 0.14 (500) |
| | 700-800 | 39.90 (-) 12.00 (100) | **44.98 (-)** 11.90 (100) | 37.13 (-) 13.63 (100) | 44.32 (-) 22.97 (96) | 33.65 (-) 0.37 (100) | 44.57 (-) 31.28 (100) | 37.79 (-) 30.01 (100) | 37.78 (-) 30.02 (100) |
| BA | 50-100 | 42.42 (0.99) 0.54 (500) | **42.43 (1.00)** 4.97 (500) | 42.05 (0.99) 2.44 (500) | **42.43 (1.00)** 0.11 (500) | **42.43 (1.00)** 0.26 (500) | **42.43 (1.00)** 0.05 (500) | **42.43 (1.00)** 0.00 (500) | **42.43 (1.00)** 0.00 (500) |
| | 700-800 | 420.89 (0.97) 3.09 (18) | **432.58 (1.00)** 5.31 (100) | 389.65 (0.90) 16.50 (80) | **432.58 (1.00)** 0.69 (100) | 432.57 (0.99) 0.28 (100) | **432.58 (1.00)** 0.06 (100) | **432.58 (1.00)** 0.01 (100) | **432.58 (1.00)** 0.02 (100) |
| HK | 50-100 | 42.32 (0.99) 0.53 (500) | **42.33 (1.00)** 5.15 (500) | 41.87 (0.98) 2.59 (500) | **42.33 (1.00)** 0.07 (500) | **42.33 (1.00)** 0.25 (500) | **42.33 (1.00)** 0.06 (500) | **42.33 (1.00)** 0.00 (500) | **42.33 (1.00)** 0.00 (500) |
| | 700-800 | 417.89 (0.98) 4.23 (19) | **429.81 (1.00)** 5.26 (100) | 390.64 (0.90) 19.66 (83) | **429.81 (1.00)** 0.42 (100) | 429.77 (0.99) 0.27 (100) | **429.81 (1.00)** 0.08 (100) | **429.81 (1.00)** 0.01 (100) | **429.81 (1.00)** 0.02 (100) |
| WS | 50-100 | 38.69 (0.99) 0.53 (500) | **38.70 (1.00)** 5.24 (500) | 37.91 (0.97) 3.80 (500) | **38.70 (1.00)** 0.07 (500) | 38.70 (0.99) 0.24 (500) | **38.70 (1.00)** 0.04 (500) | **38.70 (1.00)** 0.00 (500) | **38.70 (1.00)** 0.00 (500) |
| | 700-800 | - (-) - | **386.90 (1.00)** 5.49 (100) | - (-) - | **386.90 (1.00)** 0.39 (100) | **386.90 (1.00)** 0.28 (100) | **386.90 (1.00)** 0.05 (100) | **386.90 (1.00)** 0.00 (100) | **386.90 (1.00)** 0.01 (100) |
| HRG | 50-100 | 33.62 (0.99) 3.24 (495) | **33.72 (1.00)** 5.11 (500) | 32.80 (0.97) 5.50 (500) | **33.72 (1.00)** 0.06 (500) | **33.72 (1.00)** 0.26 (500) | **33.72 (1.00)** 0.06 (500) | **33.72 (1.00)** 0.00 (500) | **33.72 (1.00)** 0.01 (500) |
| | 700-800 | - (-) - | **304.21 (1.00)** 5.51 (100) | 221.80 (0.75) 17.81 (5) | **304.21 (1.00)** 0.44 (100) | 304.07 (0.99) 0.30 (100) | **304.21 (1.00)** 0.13 (100) | **304.21 (1.00)** 0.01 (100) | **304.21 (1.00)** 0.07 (100) |
| SATLIB | 1209-1347 | - (-) - | 426.51 (0.99) 7.99 (500) | 340.50 (0.79) 18.48 (2) | 426.55 (0.99) 13.12 (500) | 423.48 (0.99) 0.91 (500) | **426.59 (0.99)** 4.51 (500) | 426.57 (0.99) 3.12 (500) | 426.56 (0.99) 3.53 (500) |
| VC-BM | 450-1534 | 39.85 (-) 149.80 (40) | **45.10 (-)** 51.64 (40) | 37.20 (-) 139.55 (40) | 44.65 (-) 145.24 (40) | 36.02 (-) 1.95 (40) | 44.95 (-) 86.58 (40) | 42.73 (-) 266.71 (40) | 41.40 (-) 300.02 (40) |
| DIMACS | 125-4000 | 37.14 (-) 79.30 (37) | 76.86 (-) 21.26 (37) | 57.68 (-) 87.53 (37) | **79.58 (-)** 45.64 (26) | 70.11 (-) 4.13 (37) | 76.78 (-) 120.88 (37) | 74.35 (-) 202.18 (37) | 74.62 (-) 242.02 (37) |
| as-Caida | 8020-26475 | - (-) - | 19387.47 (0.99) 14.49 (122) | - (-) - | **19387.90 (1.00)** 15.63 (122) | 19387.47 (0.99) 3.65 (122) | 19387.47 (0.99) 2.23 (122) | 19387.47 (0.99) 0.15 (122) | 19387.39 (0.99) 4.60 (122) |
| PPI | 591-3480 | - (-) - | 1002.71 (0.99) 8.73 (24) | 269.00 (0.97) 23.56 (1) | 1002.50 (0.99) 3.70 (24) | 1001.54 (0.99) 1.67 (24) | **1002.83 (1.00)** 5.86 (24) | **1002.83 (1.00)** 8.16 (24) | 1001.42 (0.99) 29.02 (24) |
| REDDIT-B | 46-2283 | 146.06 (0.99) 5.94 (200) | **287.54 (1.00)** 5.37 (500) | 186.31 (0.87) 11.81 (406) | **287.54 (1.00)** 0.26 (500) | **287.54 (1.00)** 0.90 (500) | **287.54 (1.00)** 0.18 (500) | **287.54 (1.00)** 0.00 (500) | **287.54 (1.00)** 0.00 (500) |
| REDDIT-5K | 55-3648 | 194.27 (0.99) 7.97 (37) | 586.31 (0.99) 5.35 (500) | 252.74 (0.88) 15.36 (152) | **586.32 (1.00)** 0.48 (500) | 586.31 (0.99) 0.93 (500) | 586.31 (0.99) 0.22 (500) | 586.31 (0.99) 0.00 (500) | 586.31 (0.99) 0.01 (500) |
| REDDIT-12K | 41-897 | 128.61 (0.99) 3.99 (349) | **170.20 (1.00)** 5.20 (500) | 136.05 (0.92) 9.53 (467) | **170.20 (1.00)** 0.17 (500) | **170.20 (1.00)** 0.86 (500) | **170.20 (1.00)** 0.10 (500) | **170.20 (1.00)** 0.00 (500) | **170.20 (1.00)** 0.00 (500) |
| wiki-RfA | 11380 | - (-) - | **8111.00 (1.00)** 13.08 | - (-) - | 8096.00 (0.99) 19.95 | 8107.00 (0.99) 4.79 | **8111.00 (1.00)** 0.82 | **8111.00 (1.00)** 1.57 | **8111.00 (1.00)** 300.02 |
| wiki-Vote | 7115 | - (-) - | **4866.00 (1.00)** 8.56 | - (-) - | 4864.00 (0.99) 11.20 | 4864.00 (0.99) 3.94 | **4866.00 (1.00)** 0.73 | **4866.00 (1.00)** 0.87 | **4866.00 (1.00)** 300.01 |
| PubMed | 19717 | - (-) - | **15912.00 (1.00)** 12.91 | - (-) - | 15888.00 (0.99) 23.34 | **15912.00 (1.00)** 3.34 | **15912.00 (1.00)** 0.16 | **15912.00 (1.00)** 0.19 | **15912.00 (1.00)** 0.43 |
| Cora | 2708 | - (-) - | **1451.00 (1.00)** 5.63 | - (-) - | **1451.00 (1.00)** 13.63 | **1451.00 (1.00)** 2.74 | **1451.00 (1.00)** 0.06 | **1451.00 (1.00)** 0.03 | **1451.00 (1.00)** 0.11 |
| Citeseer | 3264 | - (-) - | **1808.00 (1.00)** 5.51 | - (-) - | **1808.00 (1.00)** 14.62 | **1808.00 (1.00)** 3.38 | **1808.00 (1.00)** 0.06 | **1808.00 (1.00)** 0.04 | **1808.00 (1.00)** 0.11 |
| bitcoin-alpha | 3783 | - (-) - | **2718.00 (1.00)** 6.10 | - (-) - | 2716.00 (0.99) 11.41 | **2718.00 (1.00)** 2.37 | **2718.00 (1.00)** 0.21 | **2718.00 (1.00)** 0.05 | **2718.00 (1.00)** 0.20 |
| bitcoin-otc | 5881 | - (-) - | **4346.00 (0.99)** 6.86 | - (-) - | **4352.00 (1.00)** 12.47 | 4346.00 (0.99) 2.71 | 4346.00 (0.99) 0.06 | 4346.00 (0.99) 0.08 | 4346.00 (0.99) 0.30 |
| ego-facebook | 4039 | - (-) - | **1046.00 (1.00)** 14.34 | - (-) - | - (-) - | 1034.00 (0.98) 5.50 | **1046.00 (1.00)** 19.86 | **1046.00 (1.00)** 30.05 | 1037.00 (0.99) 30.09 |
| roadnet-berlin | 61204 | - (-) - | 29843.00 (0.99) 144.22 | - (-) - | **29888.00 (1.00)** 68.22 | 29739.00 (0.99) 6.66 | 29843.00 (0.99) 1.95 | 29843.00 (0.99) 22.72 | 29843.00 (0.99) 24.01 |

# E  DATASETS

In this section, we give some details on the datasets, groups of datasets, and random graphs models used in this paper. For all datasets, we treat all graphs as undirected. The data generation module of our benchmarking suite integrates the download/generation and conversion of these graphs. We delete all self-loops, if there are any, because the solvers deal with self-loops differently (for example, with silent failures, errors, or just ignoring the vertex).

**Erdős-Rényi.**  This well-known random graph model by Erdős & Rényi (1960) connects each pair of vertices with probability $p$. We use $p = 0.15$ and for testing, generate 500 graphs with 50 to 100 vertices and 100 graphs with 700 to 800 vertices.

**Barabási-Albert.**  This random graph model by Albert & Barabási (2002) iteratively adds nodes, connecting them to $m$ already existing nodes. We use $m = 2$ and for testing, generate 500 graphs with 50 to 100 vertices and 100 graphs with 700 to 800 vertices.

**Holme-Kim.**  A random graph model by Holme & Kim (2002) similar to the BA-model, with an extra step for each randomly created edge that creates a triangle with probability $p$. We use $m = 2$ and $p = 0.05$ and for testing, generate 500 graphs with 50 to 100 vertices and 100 graphs with 700 to 800 vertices. For the large-scale experiments, we test on one graph per fixed amount of vertices, c.f. Table 4.

**Watts-Strogatz.**  A random graph model by Watts & Strogatz (1998) that starts with a well-structured ring-lattice with mean degree $k$ and in the following step replaces each edge with probability $p$ with another edge sampled uniformly at random. This way it tries to keep "small-world properties" while maintaining a random structure similar to Erdős-Rényi graphs. We use $k = 2$ and $p = 0.15$. For the large-scale experiments, we test on one graph per fixed amount of vertices, c.f. Table 4.

**Hyperbolic Random Graph.**  A random graph model by Krioukov et al. (2010), which generates graphs by randomly putting nodes in a disk in the hyperbolic plane, and connecting them if their (hyperbolic) distance is below a certain threshold. Recent works have it described to be the best model for modeling real-world networks, due to their heterogeneity and interdependency (Friedrich, 2019). As generation parameters, we use $\alpha = 0.75$, $T = 0.1$, $deg = 10$ and for testing, generate 500 graphs with 50 to 100 vertices and 100 graphs with 700 to 800 vertices. For the large-scale experiments, we test on one graph per fixed amount of vertices, c.f. Table 4.

**SATLIB.**  This is a well-known set of SAT instances in CNF commonly used as a benchmark for SAT solvers (Hoos & Stützle, 2000). A SAT instance can easily be reduced to a graph, which has a MIS as large as the number of clauses, should the SAT instance be satisfiable. In short, for each clause a clique is created, each node of which stands for a variable or its negation, e.g., $x$ or $\neg x$. Afterwards, for each variable $x$, all its nodes are connected to nodes referring to its negation $\neg x$. We use the "Random-3-SAT Instances with Controlled Backbone Size" dataset[17], which consists of 40 000 instances, of which we train on 39 500 and test on 500. Each instance has between 403 to 449 clauses.

**PPI.**  The PPI dataset, introduced by Hamilton et al. (2017), contains of 24 graphs whose nodes describe proteins and edges describe interactions between them. They contain 591 to 3 480 nodes. We use the data set provided by DGL[18], as it provides a more easily understandable separation between the graphs than the original source by GraphSAGE[19].

**REDDIT.**  The REDDIT datasets, introduced by Yanardag & Vishwanathan (2015), contain graphs constructed from online discussion threads on Reddit[20]. Vertices represent users and edges represent at least one response by one of the users to the other. The difference between REDDIT-BINARY,

---

[17] https://www.cs.ubc.ca/~hoos/SATLIB/Benchmarks/SAT/CBS/descr_CBS.html
[18] https://docs.dgl.ai/en/0.6.x/_modules/dgl/data/ppi.html
[19] http://snap.stanford.edu/graphsage/ppi.zip
[20] https://reddit.com

REDDIT-MULTI-5K, and REDDIT-MULTI-12K is how many subreddits have been crawled, and whether the classification task that is usually associated with these datasets is binary or multi-class classification. We obtain the data from TU Dortmund's data set collection[21]. For each dataset, we sample 500 graphs for testing, and they contain 41-3 648 nodes.

**as-Caida.** The as-Caida dataset of autonomous systems, introduced by Leskovec et al. (2005), consists of 122 graphs derived from a set of RouteViews BGP table snapshots. They contain 8 020 to 26 475 nodes. We obtain the data from Stanford's SNAP repository[22].

**Citation.** The Citation dataset consists of the three citation network graphs *Cora* (McCallum et al., 2000), *Citeseer* (Giles et al., 1998) and *PubMed* (Sen et al., 2008). We obtain all data from NetworkRepository (Rossi & Ahmed, 2015).

**DBLP-Coauthorship.** This dataset contains the *com-DBLP*[23] (Yang & Leskovec, 2013) graph. Vertices represent authors and an edge exists if two authors have published at least one paper together.

**Road Networks.** The Road Networks dataset contains two road networks of different sizes. The network *roadNet-PA* obtained from SNAP[24] represents the state of Pennsylvania (Leskovec et al., 2009). The other network maps the city of Berlin, Germany, and is extracted from OpenStreetMap (OpenStreetMap Contributors, 2017) using OSMnx (Boeing, 2017).

**Social Networks.** The Social Networks dataset contains of 6 social network graphs we gathered. Vertices represent users and edges represent some kind of relationship between them. The exact kind of relationship depends on the graph. The *ego-Facebook*[25] and *ego-Gplus*[26] graphs represent "friendships" between users (Leskovec & Mcauley, 2012). The *bitcoin-otc*[27] and *bitcoin-alpha*[28] graphs represent Bitcoin's web-of-trust network (Kumar et al., 2016; 2018). The *wiki-Vote*[29] and *wiki-RfA*[30] graphs are derived from interactions in the governing process of Wikipedia (Leskovec et al., 2010; West et al., 2014).

**VC-Benchmark (VC-BM).** A dataset consisting of 40 graphs of different sizes from 450 to 1 534 nodes, on which finding the maximum independent set is synthetically made hard (Xu et al., 2007). Li et al. (2018) call this dataset BUAA-MC, and sometimes it us also called BHOSLIB[31]. As the original download[32] is not available anymore, we use a GitHub mirror[33].

**DIMACS.** Graphs used for the evaluation of the second DIMACS implementation challenge[34] in 1992, on which it is particular hard to solve the MIS problem. The dataset consists of 37 graphs from 125 to 4 000 nodes. We use the complementary benchmark[35], as the original graphs were build for the CLIQUE problem.

---

[21]https://ls11-www.cs.tu-dortmund.de/staff/morris/graphkerneldatasets
[22]https://snap.stanford.edu/data/as-caida.html
[23]https://snap.stanford.edu/data/com-DBLP.html
[24]https://snap.stanford.edu/data/roadNet-PA.html
[25]https://snap.stanford.edu/data/ego-Facebook.html
[26]https://snap.stanford.edu/data/ego-Gplus.html
[27]https://snap.stanford.edu/data/soc-sign-bitcoin-otc.html
[28]https://snap.stanford.edu/data/soc-sign-bitcoin-alpha.html
[29]https://snap.stanford.edu/data/wiki-Vote.html
[30]https://suitesparse-collection-website.herokuapp.com/SNAP/wiki-RfA
[31]http://vlsicad.eecs.umich.edu/BK/Slots/cache/www.nlsde.buaa.edu.cn/~kexu/benchmarks/graph-benchmarks.htm
[32]http://www.nlsde.buaa.edu.cn/~kexu/benchmarks/graph-benchmarks.htm
[33]https://unsat.github.io/npbench/vertexcovering.html
[34]http://dimacs.rutgers.edu/archive/Challenges/
[35]http://lcs.ios.ac.cn/~caisw/graphs.html

**Amazon.** The Amazon data sets are several vehicle routing data sets provided for MWIS research (Dong et al., 2021). We test the smaller MT (6 graphs, 979-12 320 nodes), MR (5 graphs, 14 058-30 467 nodes), and MW (8 graphs, 3 079-47 504 nodes) data sets[36].

---

[36]https://registry.opendata.aws/mwis-vr-instances/

