# OpenReview forum: "What’s Wrong with Deep Learning in Tree Search for Combinatorial Optimization"
_ICLR.cc/2022/Conference — ICLR 2022 Poster_

### Official Review · Reviewer_BEvM · 2021-11-02

**Correctness:** 3
**Technical Novelty And Significance:** 1
**Empirical Novelty And Significance:** 4
**Recommendation:** 8
**Confidence:** 4

**Main Review:**

Strengths:

The empirical results of the paper have the potential to redirect (some) research in "ML for combinatorial optimization" to more fruitful directions in which it is indeed the learned components playing a vital role. I think this is fundamentally valuable.

Weaknesses:

I have multiple concerns regarding the presentation and methodology which I will list below.

* Overall presentation

First of all, Table 1 contains about 700 numbers none of which are legible on printed paper due to tiny font size (is it 3?). Given how strictly ML conferences force authors to respect font sizes, margins, and appropriate whitespace, I would almost think such a table merits a desk rejection. More importantly, even after zooming in, it is extremely hard for readers to navigate and draw meaningful conclusions from it. I suggest to a) decide on a subset of the most informative datasets (some are clearly too easy for all methods) and focus on that -- the rest can be in the appendix; b) similarly decide on the key subset of columns; surely the point isn't to compare all the heuristical components; c) This should win enough space to illustrate each of the main conclusions in a plot/table of its own (while displaying precisely the relevant information). d) Introducing a visual distinction between learning-based and classical method would be helpful.

The write-up assumes a lot of familiarity with tree-search methods and as a result, is uninviting even (!) to researchers from the wider MLxCombOpt community. I suggest the authors consider the following suggestions:

a) explaining tree-search basics in the main text

b) explaining (some of) the heuristics in the main text -- it is important for the analysis of the results anyway.

c) restructuring the related work (and introduction?) to categorize different ways ML is applied to NP-Hard combinatorial problems and explaining the place of "tree-search" in it (definitely add a discussion of learning to branch-and-bound as well as some details of (Nair 2021)). Categorizing by the degree of interaction between learned and algorithmic components is an option to consider.

With all of this in place, the paper would put itself in a position to frame its claims in much wider relevance (which I believe the claims deserve).

* Confusion about the promise of DL for comb. opt.

The second paragraph of the introduction claims that learning-based approaches give a chance to learn to solve a *specific problem* (unlike Gurobi that doesn't make a distinction). I find this inaccurate. For one, there are obviously dedicated solvers to concrete problems; this feature isn't specific to DL. But mainly, the promise of DL, as I see it, is to learn solvers specialized to a *family of instances*. This view is well-motivated by industrial reality (e.g. Amazon's routing instances are almost the same every day) and it appears in the literature (see for example the introduction to (Khalil, 2017)).

The authors should be more explicit about evaluating the methods in a "harder" setup where the learned components are expected to generalize *across* families of instances (it seemed that training happens only on SATLIB). Evaluating the more favorable setting might also be interesting.

* Treatment of Gurobi as a baseline

This paper has a unique chance to highlight a common issue with Gurobi comparisons. It has a lot of internal parameters that can be tuned to specific problems -- and in fact, they should, if other baselines are allowed to do it. Other than establishing the practice, it could further strengthen the points made by the authors.

* Clarity about GPU utilization

I assume that Gurobi and KaMIS do not utilize GPU whereas other methods do? Does rand use GPU? It seems it wouldn't need to. Given the massive computational advantage that 8 V100s give, the runtime comparisons should be very clear about it. Again, this clarity should go in favor of the main message: e.g. "even with such computational advantage tree-search doesn't outperform traditional solvers" or "since tree search is comparable to rand; it is basically just a powerful GPU-friendly heuristical algorithm that is independent of any machine learning".

**Summary Of The Paper:**

The paper conducts a thorough experimental evaluation of a line of work on "deep-learning guided tree search in combinatorial optimization". This line of work iterates on a relatively generic greedy-style algorithm for NP-Hard graph optimization problems; with a neural network -- trained on a training set of instances -- guiding the greedy decisions. The popularity of this approach is anchored in the premise of "deep learning will help us solve NP-Hard problems.

Findings of the experimental evaluation are the following:
1) Some earlier work is not reproducible
2) Traditional dedicated solvers are comparable or better; particularly on harder/larger problem instances
3) Performance of data-driven methods barely changes when outputs of trained neural nets are replaced by random values (i.e. all the algorithmic power comes from explicit and traditional heuristical components)

**Summary Of The Review:**

Despite the paper being purely experimental, its main point can have significant net-positive impact within the wider research area. Not only by casting doubt on the entire "DL based tree-search" but also by serving as a long-term warning -- claims about "outperforming SOTA of classical combinatorial optimization" can fall apart under proper experimental methodology.

 For this reason, I am comfortable disregarding the usual demands for technical novelty, beating benchmarks, or providing theory.

However, in its current form, I do not believe the paper would realize its potential impact due to the issues listed above. I believe they require significant changes in the paper structure so I cannot, at this point, recommend acceptance.

---

> ### Author Response · Authors · 2021-11-12
> **Response to Reviewer BEvM**
>
> Thank you very much for your very thorough and insightful review. We would like to address your points individually.
>
> ### Overall presentation
> We received your criticism of the presentation and tried to improve it following your and the other reviewers’ feedback. We’ve moved the two tables into the appendix and included a small subset table for the main text. We’ve further structured the evaluation text more clearly by labeling the paragraphs, and added some extra plots.
>
> Regarding the visual distinction, for the original table 1, this just were learning-based methods, and in the original table 2, we compared the full/default configurations of them to the classical solvers. Maybe this was a bit confusing, but as it just was 1-2 column(s) per solver in table 2, we hope it is sufficiently clear.
>
> > The write-up assumes a lot of familiarity with tree-search methods
>
> We agree that the write-up assumes a lot of familiarity, and hence moved the textual description of the algorithm into the main text. Due to space constraints, the pseudo code has to stay within the Appendix. Unfortunately, the heuristics needed to stay in the appendix, to make some space for the plots that reviewer m13R suggested. We hope that using the text (and maybe a quick look into the pseudocode), the entry barrier for the paper is now a bit lower.
>
> ### Related Work Restructuring
> Thank you very much for that suggestion! Indeed, the related work discussion was not very strong until now. We’ve integrated the related work section into section 2, and while doing that tried to discuss the different variants on how the algorithmic components and algorithms can interact and what the design space is (including branch-and-bound as well as Nair 2021). We hope that we now provide a much clearer introduction into the current state of the art. If you have any further suggestions for that, we are willing to include them for a final version.
>
> ### Confusion about the promise of DL for comb. opt.
> We agree we were confusing learning problem-specific solution structure and instance-specific solution structure. Thank you very much for pointing that out. We’ve tried to make that more clear in the revised version, both in the introduction and the evaluation, where we discuss that the DGL/Intel-Treesearch neither learn the instance-specific structure (SATLIB) nor generalize beyond the training family.
>
> ### Gurobi baseline tuning
> While we think it’s not part of our main paper to evaluate parameter tuning for Gurobi, we agree it’s an interesting experiment. Using Gurobi’s internal optimization tool `grbtune`, we added another appendix section that discusses the impact of tuning Gurobi. For our use case, it shows that parameter tuning does not bring additional performance and Gurobi’s defaults are sensible.
>
> ### Clarity about GPU utilization
> Indeed, Gurobi and KaMIS are CPU-only, just as the randomized DGL-TreeSearch. Note that in the experiments we just used one GPU per experiment because even the multithreaded tree search was not able to even get close to maximum GPU utilization. Additionally, we ran a multithreaded tree search with 8 GPUs and see no impact on performance (which is explained by the non-GPU-boundedness of the tree search). We’ve tried to make this more clear in the revised paper.
>
> Thank you very much again for your feedback. We hope we addressed your points accordingly, and restructured the paper in a way that you now can comfortably make an acceptance decision. We are open for further feedback.
>
> Kind regards,
> Authors

---

> > ### Comment · Reviewer_BEvM · 2021-11-21
> > **Good rebuttal**
> >
> > I thank the authors for a thorough rebuttal.
> >
> > The paper is now definitely above the acceptance threshold.
> >
> > To be honest, I still think the paper undersells itself and is tricky to linearly navigate for anyone but most specialized readers. But the new version is a clear improvement and given the overall volume of the material and the overall complexity of the setup, one has to admit that a neat and crisp write-up is a Herculian task.
> >
> > My updated score would be a clear 7 but this option is currently not available. Unless it opens up later in the review process, may the authors enjoy the rounded-up rating :).

---

> > > ### Author Response · Authors · 2021-11-22
> > > **Thank you very much for your reply.**
> > >
> > > Thank you very much for your positive reply and the score increase. We are very happy that you enjoy the revised paper and again want to thank you for the helpful feedback.
> > >
> > > If you have any further hints on how to fix the write-up to make the paper more accessible, we would be glad to incorporate them for a possible final version.
> > >
> > > Again, thank you so much for your time and the review, which greatly improved the paper.

---

### Official Review · Reviewer_m13R · 2021-11-02

**Correctness:** 4
**Technical Novelty And Significance:** 1
**Empirical Novelty And Significance:** 3
**Recommendation:** 8
**Confidence:** 5

**Main Review:**

**Strengths**

- Clarity: The paper is generally very well-written, although the presentation of the results can be improved substantially.
- Motivation: Li et al.’s paper is being cited and compared against in tens of papers yearly, and so understanding its limitations and apparently fundamental issues is useful for the community.
- Reproducibility: The authors are very systematic and transparent in how they generate datasets, implement the various methods, and evaluate them. I can see their code becoming widely used and built on in the development of ML-based methods for MIS.

**Weaknesses**

1. Presentation of experimental results: Tables 1 and 2 should be part of the paper’s appendix for sure, but you really need to find better ways of presenting those thousands of statistics in the main text. As things currently stand, the reader needs to zoom-in to read the numbers; there are so many columns that it’s hard to track which one corresponds to which method or what the trends are in terms of best method for a given dataset. Additionally, please label the paragraphs of section 3.1 so that the reader can immediately understand which aspect of the results you’re discussing. As things stand, I’ve had to decipher which columns I should query in the tables to see what you’re saying in the paragraphs of 3.1.

2. Statistical metrics: The average optimality ratios and running times in Tables 1 and 2 are certainly indicative of some trends. However, some box plots of the distributions of the optimality ratios/running times might shed more light into the robustness of the methods. For instance, the running time average may be biased by outliers whereas a box plot factors that in. Combining this suggestion with the one above, you could consider moving the tables to the appendix and replacing them in the main text with two box plots per datasets, one for optimality ratio and another for running time. This way, the reader can visually compare different methods without having to zoom-in and read hundreds of numbers. Since this paper’s contributions are largely software/empirical results, you can also consider performing statistical testing for each pair of methods; see the two-sided Wilcoxon Signed Rank Test for example.

3. Datasets: Please consider additional datasets which may be a bit more standard for MIS papers, e.g.:
- http://vlsicad.eecs.umich.edu/BK/Slots/cache/www.nlsde.buaa.edu.cn/~kexu/benchmarks/graph-benchmarks.htm
- http://lcs.ios.ac.cn/~caisw/graphs.html

In particular, the DIMACS implementation challenge graphs have been used in the KaMIS paper for example, among others. Also, this very recent dataset of large-scale instances may be of interest (even if only to evaluate and not train): https://arxiv.org/abs/2105.12623

4. MIS heuristics and mathematical programming formulations: Please check Butenko’s dissertation (Butenko, Sergiy. Maximum independent set and related problems, with applications. University of Florida, 2003.), particularly chapters 2-3 and the experimental results later on. The binary linear programming formulation you used with Gurobi is not the only one possible; there are quadratic formulations (see eq. (2.3) in Butenko) which may be easier to solve in practice than the linear one. Also please check the famous GRASP heuristic for MIS and consider implementing it: Feo, Thomas A., Mauricio GC Resende, and Stuart H. Smith. "A greedy randomized adaptive search procedure for maximum independent set." Operations Research 42.5 (1994): 860-878.

5. Where do your results leave us? I would’ve expected you to identify datasets (existing or new) for which KaMIS and Gurobi underperform in some respect. Perhaps that is tricky; you’ve tried many datasets (though there are more you could try as mentioned earlier) and the two solvers did very well. You might then want to consider other harder versions of MIS, for example the Generalized Independent Set Problem, see (Colombi, Marco, Renata Mansini, and Martin Savelsbergh. "The generalized independent set problem: Polyhedral analysis and solution approaches." European Journal of Operational Research 260.1 (2017): 41-55.) and (Hosseinian, Seyedmohammadhossein, and Sergiy Butenko. "Algorithms for the generalized independent set problem based on a quadratic optimization approach." Optimization Letters 13.6 (2019): 1211-1222.). In this variant, some edges may be “purchased” and their endpoints may violate the independence requirement. This makes the problem much harder than MIS for integer programming solvers. Your paper should really push the community to advance the field.

6. Solver parameter tuning as a baseline: Please consider adding a baseline in which KaMIS/Gurobi are “trained” on the same datasets as the ML-based methods by tuning their parameters using off-the-shelf tools like SMAC (https://www.automl.org/automated-algorithm-design/algorithm-configuration/smac/). Such a baseline combines the best of both worlds in a sense: the stability and generality of these solvers with the potential benefits of leveraging the instance distribution.

Minor comments:

- page 8, “Overall, we see that … cannot deal with some graphs”, this is not true for Gurobi though, correct?

**Summary Of The Paper:**

This paper proposes a software package for generating data and training/evaluation some ML and non-ML approaches for the Maximum Independent Set (MIS) problem in its both its unweighted and weighted versions. It is shown that a highly-cited method that combines supervised learning using a graph neural network (GCN) with a (complete) tree search does not actually need the GCN if the MIS instance is preprocessed appropriately using existing non-ML based codes. On the other hand, a recent deep reinforcement learning (DRL) approach for the same problem is shown to actually use its GCN’s predictions, obtaining better results overall. Last, non-ML solvers such as KaMIS and Gurobi are shown to find better solutions in a short amount of time for most instance datasets, putting into question the potential for ML-based approaches in general for MIS.

**Summary Of The Review:**

Overall, I like this paper and think it makes a solid contribution to the intersection between deep learning and combinatorial optimization. However, I think a paper that “debunks” a highly-cited work should also establish convincing avenues for future research which are currently beyond the scope of (ML or non-ML) existing methods; I argue that this is missing at this stage. I would like to see the authors’ responses to my questions and concerns before making a final decision, but am generally positive about this submission.

---

> ### Author Response · Authors · 2021-11-12
> **Response to Reviewer m13R**
>
> Thank you very much for your very thorough and insightful review. We would like to address your points individually.
>
> ### Presentation and Metrics
> As you’ve suggested, we moved the long tables into the appendix and included a much smaller table for the main tree search results, focussing on two random graphs (ER, HRG), and a few real-world data sets. Furthermore, we’ve followed your suggestion to label the paragraphs of the analysis. Thank you very much!
>
> We’ve added violin plots for some data sets comparing the different configurations, both for the tree searches and the other solvers. We think violin plots are more suitable than boxplots for our purposes here, and agree this is a valuable addition to the paper.
>
> ### Datasets
> We’ve added the DIMACS implementation challenge to our evaluation. Thank you for making us aware. The other link and resources on the first link point to "BHOSLIB" which is equivalent to VC-BM in our paper. We added that name to the VC-BM subsection, to avoid future confusion.
> We’ve also included the Amazon MWIS dataset in our evaluation, at least on the smaller instances, as all solvers except Gurobi run into their limits on these graphs.
>
> ### MIS heuristics
> Although we believe evaluating different variants for solving MIS with Gurobi is not the main focus of the paper, we added an additional appendix section that compares the linear formulation with the quadratic.
>
> Regarding the GRASP heuristic, we researched that algorithm and could not find any recent modern implementation (just a Fortran one), so we believe KaMIS to be the more widespread and modern heuristic solver. If you can point us to a modern (Python or C(++)) implementation, we would be willing to integrate it into the analysis, but we are not able to implement this algorithm within the timeframe for the discussion period in a way that would be fair, as we do not want to have some prototypical non-tested implementation that was built in a day. We hope you understand our decision here, and thank you very much for the pointer. In any case, we added GRASP to the related work section.
>
> ### Generalized Independent Set
> We agree that such a paper should push the community to advance the field and believe it would be a worthwhile addition to consider other versions of MIS (note that we already analyze the weighted variant, which makes the problem much harder, as vertices are not valued uniformly anymore). However, the related work we could find within the ML4CombOpt community does not target different MIS variants. With our paper, we wanted to make a first step towards a reproducible evaluation of the state of the art, and we are already at the limits of our conference submission. We agree that for a longer journal paper, it would be interesting to apply some solvers to other problems, but this would imply further engineering and we believe this to be out of scope for this paper. We added a discussion of this to the future work section and hope that our paper motivates the field to continue to push forward with a focus on rigorous empirical evaluation.
>
> ### Gurobi tuning
> While we think it’s not part of our main paper to evaluate parameter tuning for Gurobi, we agree it’s an interesting experiment. Using `grbtune`, we added another appendix section that discusses the impact of tuning Gurobi. For our use case, it shows that parameter tuning does not bring additional performance and Gurobi’s defaults are sensible.
>
> Regarding your minor comment, our formulation was a bit confusing as “algorithmic solvers” was referring to KaMIS only. We made it more clear in the revised version.
>
> After addressing the technical problems you mentioned as best as we could within the limited timeframe of the discussion period, the question still stands whether a “debunking” paper should establish conving avenues for future research, as you write. With this paper, it is our goal to show how important rigorous empirical evaluation of newly proposed methods, especially in a rapidly evolving field such as machine learning, is. While we agree that for example other MIS variants would be interesting, we believe that taking the step back and check whether the assumption “using ML, we can efficiently solve MIS” as Li et al. are often cited is correct, already advances the field a lot and lays the ground for future research on these methods. A formal analysis on why it actually is difficult to learn a solution structure here would be very insightful for future work. For such a “debunking” benchmarking paper, we believe it’s out of scope to additionally propose a novel method that then the next steps build up upon. But we understand your point and hope that our improvements, clarifications and additional evaluations will guide your tree search for an acceptance decision :-).
>
> Thank you very much again, we really believe your feedback to have greatly improved our paper. Please let us know if this addresses your concerns for a final decision.
>
> Kind regards,
> Authors

---

> > ### Comment · Reviewer_m13R · 2021-11-18
> > **Response to response**
> >
> > Presentation and Metrics
> >
> > Great! Consider adding median and lower/upper quartile lines to the violins, so that they provide strictly more information than their box plot counterparts. It would be good if the reader can immediately identify the median MIS size without having to visually estimate it by integrating the density.
> >
> > Datasets
> >
> > Great!
> >
> > MIS heuristics
> >
> > Understood. Can you confirm that you use the branch and reduce algorithm in KaMIS as in section 4.3 of the user guide (http://algo2.iti.kit.edu/schulz/software_releases/kamis.pdf)? Seems like it from looking at your code, but it would be good to clarify to the reader.
> >
> > Generalized Independent Set
> >
> > Understood.
> >
> > Gurobi tuning
> >
> > Well done, thank you.
> >
> > Overall comment: this is an excellent rebuttal. I will increase my evaluation from 6 to 7, but no more. Your finding that LwD does better than the supervised methods is not necessarily new in that the LwD paper already makes that case. The "debunking" part is interesting in its own right, as is the contribution of a complete codebase for learning in MWIS with Tree Search, hence the increase.

---

> > > ### Author Response · Authors · 2021-11-22
> > > **Thank you for your answer.**
> > >
> > > Thank you very much for your positive reply and the score increase. We are very happy that you enjoy the revised paper and again want to thank you for the helpful feedback.
> > >
> > > We will update the violins for a possible final version of the paper.
> > >
> > > We use the section 4.3 weighted branch and reduce algorithm for the weighted case, and ReduMIS as described in section 4.1 of the user guide for unweighted graphs. Note that you could also input a "weighted" graphs with all weights 1 into the weighted branch and reduce algorithm, however, ReduMIS performs better for these cases, because it uses specialized reductions for the unweighted case.
> > >
> > > Again, thank you so much for your review and time.

---

### Official Review · Reviewer_oh3B · 2021-11-03

**Correctness:** 3
**Technical Novelty And Significance:** 2
**Empirical Novelty And Significance:** 3
**Recommendation:** 6
**Confidence:** 4

**Main Review:**

Strengths:
- I think the paper addresses interesting and important questions. Understanding what work and what does not work, and which parts of a solution actually contribute to the performance, is important.
- The paper provides interesting insight on the performance and reproducibility of a previous work (Li et al. [2018]) based on thorough experiments.
- The paper presents a new benchmark suite for MIS that includes a large number of benchmark instances and implementations of several popular approaches.

Weaknesses:
- All experiments are done on one problem type, MIS, while there is a lot of work on other graph-related problems such as TSP, VRP, etc. For example, Li et al. [2018] that is discussed in this work have considered other problems. It is hard to draw conclusions on combinatorial optimization from one problem.
- The paper focuses on a single work (Li et al., 2018) that the authors were unable to reproduce and a single work that showed promising results (Ahn et al. [2020]). A study of a larger sample of deep learning solutions would be useful to support claims about the value of GNNs, neural-guided tree search, or reinforcement learning for combinatorial optimization.
- The result that specialized solvers and even classical solvers are often better than deep learning solutions, especially on larger problems, has been reported for other computational problems (e.g., TSP [1]). Ahn et al. [2020] already reported that KaMIS outperforms Li et al. [2018] and that reduction and local search lead to improvements. Further, other works have looked at generalization of GNNs (e.g., [2]). These need to be cited and the similarities and differences with this work should be discussed.
- While providing open-source benchmark suite, including implementations of several popular approaches, is important, as far as I understand this suite is primarily a collection of existing problems and randomly-generated graphs and does not introduce new benchmark datasets. I am not sure this is an important contribution of the work.



[1] Joshi, C. K., Cappart, Q., Rousseau, L. M., Laurent, T., & Bresson, X. (2020). Learning TSP requires rethinking generalization. arXiv preprint arXiv:2006.07054.

[2] Xu, K., Zhang, M., Li, J., Du, S. S., Kawarabayashi, K. I., & Jegelka, S. (2020). How neural networks extrapolate: From feedforward to graph neural networks. arXiv preprint arXiv:2009.11848.


**Summary Of The Paper:**

The paper presents an evaluation of deep learning-based tree serch solutions (that are based on graph neural networks) for solving combinatorial optimization problems. They present an open-source benchmark suite for the maximum independent set (MIS) problem (both weighted and unweighted) that includes instances from multiple random graph models, known benchmark suites (e.g., SATLIB) and other graphs from the literature. They conduct experiments on different configurations of neural-guided tree search and show that the results by Li et al. [2018] are not reproducible. They also show that general and tailored classical solvers outperform deep learning solutions.

**Summary Of The Review:**

Overall, I think this type of works is important and can lead to important insight. However, I think experiments with more problem types and more solutions are needed to draw general and interesting conclusions. Also, the paper needs to discuss some relevant works that are currently not mentioned.

---

> ### Author Response · Authors · 2021-11-12
> **Response to Reviewer oh3B**
>
> Thank you very much for your thorough and insightful review. We would like to address your points individually.
>
> > All experiments are done on one problem type, MIS, while there is a lot of work on other graph-related problems such as TSP, VRP, etc.
>
> We agree that for a final verdict on general combinatorial optimization, other graph problems like the ones that you mention should be evaluated. However, the weighted variant of MIS that we consider already makes the problem a lot harder, as vertices are not valued uniformly anymore. While Li et al. consider other problems, they are not vastly different at their core (e.g., Minimum Vertex Cover is just a flipped Maximum Independent Set), compared to TSP/VRP vs MIS. Due to time and space constraints we are not able to include detailed analyses of TSP/VRP architectures and we believe that it would be a bit out of scope for this paper; however, we made it more clear in the paper why we focus on MIS and added to the conclusion that future analyses on other problems, like TSP, should be made. Additionally, in Section 2, we additionally discus different architectures for ML-based combinatorial optimization, and discussed the TSP architecture by Kool et al. [2].
>
> > A study of a larger sample of deep learning solutions would be useful to support claims about the value of GNNs, neural-guided tree search, or reinforcement learning for combinatorial optimization.
>
> We agree that adding more solvers to the benchmarking suite and analysis would provide more insights. However, the most important claims we make in the paper target specifically the Intel-TreeSearch/DGL-TreeSearch. It is not our goal to evaluate all approaches of DL4CompOpt that are out there; we believe this would be more fitting for a longer journal article, as space and scope of a conference submission are limited. Furthermore, we are not certain which solvers we would be missing. Notably, there is S2V-DQN [1] as a reinforcement learning solver, but Learning what to Defer has proven to be better than it in every aspect, so it is not included in our analysis due to space constraints and we have more space for the detailed tree search analysis.
>
> > The result that specialized solvers and even classical solvers are often better than deep learning solutions, especially on larger problems, has been reported for other computational problems
>
> Thank you very much for making us aware of these papers. During the restructuring of the related work discussions that another reviewer suggested, we have added them to our related work and evaluation discussions and made it more clear that Ahn et al. in the LwD paper also analyze KaMIS.
>
> > While providing open-source benchmark suite, including implementations of several popular approaches, is important, as far as I understand this suite is primarily a collection of existing problems and randomly-generated graphs and does not introduce new benchmark datasets. I am not sure this is an important contribution of the work.
>
> Our suite provides a very comprehensive collection of standard real-world, random, and synthetically made hard graph instances on which the MIS problem is solved. We think the core contribution of a benchmarking suite is exactly that: Creating such a central collection of datasets, unifying the solvers -- which often are just available in a “research-code” status -- under a single interface, and providing the results. We do not think that introducing a new MIS benchmarking dataset is within the scope of our paper, especially because focusing on the existing ones enables comparison with other works. Often, there are papers published just for the release of a single new benchmark dataset. Based on another reviewer’s feedback, we added the DIMACS and Amazon datasets to the paper.
>
> Again, thank you for helping to improve the paper! We hoped we addressed the points that we believe to be within the scope of this conference submission, and made it clear why we think that targeting more problems like TSP does not fit into the scope of this submission. Furthermore, if there is any other solver you are missing that would provide additional insights, we would be grateful for a pointer. Please let us know if this addresses your concerns.
>
> Kind regards,
> Authors
>
> [1] Elias B. Khalil, Hanjun Dai, Yuyu Zhang, Bistra Dilkina, and Le Song.  Learning combinatorial optimization algorithms over graphs. In: Advances in Neural Information Processing Systems(NeurIPS), volume 30, 2017.
>
> [2] Wouter Kool, Herke van Hoof, and Max Welling. Attention, learn to solve routing problems!  In: Proceedings of the 7th International Conference on Learning Representations (ICLR), 2019

---

> > ### Comment · Reviewer_oh3B · 2021-11-30
> > **Thank you for the response**
> >
> > I thank the authors for their response and for revising the paper to address the concerns. I have increased my score from 5 to 6.

---

### Official Review · Reviewer_Qd1X · 2021-11-08

**Correctness:** 3
**Technical Novelty And Significance:** 3
**Empirical Novelty And Significance:** 3
**Recommendation:** 8
**Confidence:** 3

**Main Review:**

The primary strength of the paper is it is opening up an important direction towards scientific accuracy. Citation counts often work as a proxy for the perceived importance of a paper. A paper with 200+ citations in less than 3 years of publication is thus likely to be considered as highly influential in the field. Through re-looking at the claimed results, this paper makes a significant contribution towards a research philosophy that seeks to validate influential papers.

The benchmark data set is an important contribution and the detailed results presented in Table 2 can immensely benefit future performance comparisons.

The writing of the paper is excellent and the paper and the literature review is extensive.

That said, one potential risk in this paper is what if the current implementation is wrong and indeed the guided tree search is effective?

**Summary Of The Paper:**

The paper looks at an important problem: combinatorial optimization that arises in several real-world settings. Next, the paper focuses on a influential paper in the field (Li et al. 2018, 200+ citations so far) that presents a GNN approach and reports impressive performance numbers. Next, the paper presents investigates if these results are reproducible by (1) running the publicly available implementation after fixes; and (2) re-implementing the algorithm as described in the paper and documentation. None of these versions replicates the reported performance. The paper also presents a benchmark suite to make comparison of this task easier.

**Summary Of The Review:**

Our field is producing papers at a fast rate. Reviewers can provide scientific checks and balances only up to a certain level. This type of rigorous effort can offer valuable information about highly influential papers to the scientific community.

---

> ### Author Response · Authors · 2021-11-12
> **Response to Reviewer Qd1X**
>
> Thank you very much for your very positive review and feedback. Regarding your point that one potential risk is that the guided tree search might actually be effective, we agree that there is no formal proof of correctness of the code, but as we observe similar behavior on the original implementation as well as our implementation from scratch, both with the provided weights as well as with newly trained weights, we believe that the probability for the error lying within a faulty implementation is rather small.
>
> Other than that, again, thank you very much for your feedback.
>
> Kind regards,
> Authors

---

### Author Response · Authors · 2021-11-12
**Updated manuscript with revised structure and additional appendix sections**

We would like to thank all of the reviewers again for helping us improve the paper. We just uploaded a revised version of our paper, and answered your individual suggestions within your comments. In a nutshell, we

- restructured the related work/background section, explaining the design space of deep learning for combinatorial optimization
- improved the presentation of our data by just showing a smaller representative table in the main text and moving the large tables into the appendix
- moved some explanation of the tree search algorithm into the main text
- motivated DL for combinatorial optimization clearer (learning families of instances)
- structured the evaluation texts better
- added violin plots to analyze the robustness of the methods
- added the DIMACS implementation challenge graphs and Amazon MWIS instances to our datasets
- added a quadratic variant for formulating the MIS in Gurobi  to an additional evaluation section in the appendix
- analyzed the impact of Gurobi parameter tuning using grbtune in an additional evaluation section in the appendix
- fixed minor details

Thank you all again for your very valuable and insightful suggestions. Please let us know if you have additional questions or ideas for improvement.

Kind regards,
Authors

---

### Decision · Program_Chairs · 2022-01-20

**Decision:**

Accept (Poster)

**Comment:**

I would like to thank the authors for having managed a thorough discussion despite the complexity of the task at hand (e.g. BEvM). during discussion, the reviewers clearly converged to accepting the paper, praising the importance of the problem tackled and the setup put in place to effectively tackle the challenge at hand.

All this makes the paper an important contribution and a clear accept (and an enjoyable read), for which I can only recommend a further polish before camera ready to follow the latest inclusions.

AC.